# Exponentially Improving the Complexity of Simulating the Weisfeiler-Lehman Test with Graph Neural Networks

**Anders Aamand**
MIT
aamand@mit.edu

**Justin Y. Chen**
MIT
justc@mit.edu

**Piotr Indyk**
MIT
indyk@mit.edu

**Shyam Narayanan**
MIT
shyamsn@mit.edu

**Ronitt Rubinfeld**
MIT
ronitt@mit.edu

**Nicholas Schiefer**
MIT
schiefer@mit.edu

**Sandeep Silwal**
MIT
silwal@mit.edu

**Tal Wagner**[*]
Amazon AWS
tal.wagner@gmail.com

## Abstract

Recent work shows that the expressive power of Graph Neural Networks (GNNs) in distinguishing non-isomorphic graphs is exactly the same as that of the Weisfeiler-Lehman (WL) graph test. In particular, they show that the WL test can be simulated by GNNs. However, those simulations involve neural networks for the "combine" function of size polynomial or even exponential in the number of graph nodes $n$, as well as feature vectors of length linear in $n$.

We present an improved simulation of the WL test on GNNs with *exponentially* lower complexity. In particular, the neural network implementing the combine function in each node has only $\mathrm{polylog}(n)$ parameters, and the feature vectors exchanged by the nodes of GNN consists of only $O(\log n)$ bits. We also give logarithmic lower bounds for the feature vector length and the size of the neural networks, showing the (near)-optimality of our construction.

## 1 Introduction

Graph Neural Networks (GNNs) have become a popular tool for machine learning on graph-structured data, with applications in social network prediction [HYL17], traffic prediction [YYZ18], recommender systems [YHC+18], drug discovery [WKK+20], computer vision [LGD+19, FLM+19, MBM+17, QSMG17], and combinatorial optimization [CCK+21]. Standard message passing GNNs use the topology of the input graph to define the network structure: in each step $k$, a node aggregates messages from each of its neighbors and combines them using a function $\phi^{(k)}$, computed by a neural network, to determine its message for the next round. Crucially, the aggregation function must be *symmetric*, to ensure that the output of GNNs is invariant under node permutation. This restriction raised questions about how expressive such network architectures are, and in particular what classes of graphs are distinguishable using GNNs.

The seminal works of Xu et al. [XHLJ19] and Morris et al. [MRF+19] (see also [Gro21]) showed that GNNs are exactly as powerful in distinguishing graphs as the Weisfeiler-Lehman (WL) test [WL68], also known as *color refinement*. This combinatorial procedure is a necessary but not sufficient test for graph isomorphism. It proceeds in repeated rounds: in each round, a node labels itself with the "hash" of the multiset of labels of its neighbors. The aforementioned papers show that (i) GNNs can simulate the WL test and (ii) GNNs can only distinguish those graphs that the WL test determines to be different. This provides a complete characterization of the expressive power of GNN architectures.

---

[*]Work done prior to joining Amazon.

36th Conference on Neural Information Processing Systems (NeurIPS 2022).

Table 1: Our results compared to prior work on GNNs that simulate the WL test.

| Construction | Message size | Parameters in $\phi^{(k)}$ | Oblivious[*] | Deterministic[*] |
|---|---|---|---|---|
| Xu et al. [XHLJ19] | $O(n)$ | $\Omega(2^n)$ | Yes | Yes |
| Morris et al. [MRF+19] | $O(n)$ | $\mathrm{poly}(n)$ | No | Yes |
| This paper | $O(\log n)$ | $\mathrm{polylog}(n)$ | Yes | No |

[*] Oblivious designs use the same weights for any input graph. Non-deterministic constructions require some weights to be assigned by random samples drawn from some distribution, and may err with small probability.

The connection between GNNs and the WL test has spawned a wave of new results studying GNN variants that either match the distinguishing power of the WL test or adopt new methods beyond message passing on the edges of the input graph to overcome this barrier (see the excellent surveys [MFK21, MLM+21, Gro21] for an overview of this area). However, the results in the original as well the follow up works mostly focus on *qualitative* questions (how expressive GNNs are) as opposed to *quantitative* questions such as the network complexity. In particular, while Xu et al. [XHLJ19] show that there exist GNN architectures that can simulate the WL coloring procedure as long as the aggregation step is injective, they rely on the universal approximation theorem to show that there exists a neural network that can simulate the hash function used in WL. As a result, the size of the network could be exponential in the number of nodes $n$. In contrast, the construction of Morris et al. [MFK21] uses networks of size polynomial in $n$. However, the weights of the network implementing $\phi^{(k)}$ in their construction depend on the structure of the underlying graph, which suffices for *node* classification, but is not sufficient for the context of *graph* classification.

Overall, the quantitative understanding of the complexity of simulating WL remains an open problem. Indeed, the survey [Gro21] states *"The size of the GNNs and related parameters like depth and width, which directly affect the complexity of inference and learning, definitely require close attention."*

**Our Results.** The main question addressed in this work is: what is the simplest (in terms of the number of neural network units and message length) GNN capable of simulating the WL test? Equivalently, at what point does a GNN become so small that it loses its expressive power?

Our main result is a highly efficient construction of a GNN architecture that is capable of simulating the WL test. For graphs with $n$ nodes, it can simulate $\mathrm{poly}(n)$ steps of the WL test, such that the neural network implementing $\phi^{(k)}$ in each round has $\mathrm{polylog}(n)$ parameters, and the messages exchanged by the nodes of the GNN in each round consist of $O(\log n)$ bits. This offers at least an exponential improvement over the prior bounds obtained in [XHLJ19, MRF+19] (see Table 1), extending the equivalence between the WL test and the expressive power of GNNs to neural networks of reasonable (in fact, quite small) size. Furthermore, our architecture is simple, using vector sum for aggregation and ReLU units for the combine function $\phi^{(k)}$. Finally, our construction can be generalized to yield a depth-size tradeoff: for any integer $t > 0$, we can can construct a neural network of depth $O(t)$ and size $n^{O(1/t)} \mathrm{polylog}(n)$.

To achieve this result, our construction is randomized, i.e., some weights of the neural networks are selected at random, and the simulation of the WL test is correct with high probability $1 - 1/\mathrm{poly}(n)$. Thus, our construction can be viewed as creating a *distribution* over neural networks computing the function $\phi^{(k)}$.[2] In particular, this implies that, for each graph, there exists a single neural network implementing $\phi^{(k)}$ that accurately simulates WL on that graph. The size of the network is exponentially smaller than in [MRF+19], although the construction is probabilistic.

We complement this results with two lower bounds for executing a WL iteration. Our first lower bound addresses the *communication complexity* of this problem, and demonstrates that to solve it, each node must communicate labels that are at least $O(\log n)$ bits long, matching the upper bound achieved by our construction. Our second lower bound addresses the *computational complexity*,

---

[2]Note that selecting $\phi^{(k)}$ at random is quite different from random node initialization, e.g., as investigated in [ACGL21]. In particular, in our model all nodes use *the same* function $\phi^{(k)}$ (with the same parameters), without breaking the permutation invariance property of GNNs, as in the standard GNN model.

namely the parameters of the neural network. It shows that if the messages sent between nodes are vectors with entries in $[F] = \{0, 1, \ldots, F - 1\}$, then the network implementing $\phi^{(k)}$ must use $\Omega(\log F)$ ReLU units.

**Related work.** The equivalence between the discriminative power of GNNs and the WL test has been shown in the aforementioned works [XHLJ19, MRF$^+$19]. A strengthened version of the theorem of [XHLJ19], where the same combine function $\phi$ is used in all iterations (i.e., $\phi = \phi^{(k)}$ for all $k$) appeared in the survey [Gro21]. Many works since have studied various representational issues in GNNs; we refer the reader to excellent surveys [Gro21, HV21, MLM$^+$21, Jeg22]. In particular, [Lou19] established connections between GNNs and distributed computing models such as LOCAL and CONGEST, and derived lower bounds for several computational tasks based on this connection. [CVCB19] drew a connection between the expressiveness of GNNs in graph isomorphism testing and in function approximation. [BKM$^+$20] studied the expressiveness of GNNs in computing Boolean node classifiers, and [GMP21] studied the expressiveness of graph convolutional networks (GCNs).

The emergence of WL as a barrier in GNN expressivity has also led to a flurry of work on enhancing their expressivity by means of more general architectures. These include higher-order GNNs inspired by higher-dimensional analogs of WL [MRF$^+$19, MBHSL19], unique node identifiers [Lou19, VLF20], random node initializations [ACGL21, SYK21], relational pooling [MSRR19], incorporating additional information on the graph structure [NM20, BGRR21, BFZB22, CMR21, TRWG21], and more. We refer to [MLM$^+$21] for a comprehensive survey of this line of work.

## 1.1 Preliminaries

**Notation.** For the rest of the paper, we use $\mathcal{N}(v)$ to denote the neighborhood of $v$ in a graph $G(V, E)$ *including $v$ itself*, and we use $\{\cdot\}$ to denote *multisets* rather than sets.

**GNNs.** Let $G = (V, E)$ be a graph with $N$ nodes. GNNs use the graph structure of $G$ to learn node embeddings for all the nodes across multiple iterations. Let $h_v^{(k)}$ denote the embedding vector of node $v \in V$ in the $k$th iteration. The vectors $h_v^{(0)}$ represent the initial node embeddings. In every iteration $k \geq 1$, each node $v$ sends its current embedding $h_v^{(k-1)}$ to all its neighbors, and then computes its new embedding $h_v^k$ by the equation

$$h_v^{(k)} = \phi^{(k)} \left( f \left( \left\{ h_w^{(k-1)} : w \in \mathcal{N}(v) \right\} \right), h_v^{(k-1)} \right) \tag{1.1}$$

where $\phi^{(k)}$ is implemented by a neural network with ReLU activations (note that $\phi^{(k)}$ may differ across different iterations $k$). The function $f$ is called the 'aggregate' function, and $\phi^{(k)}$ is called the 'combine' function. The embeddings $h_v^{(K)}$ at the final iteration $K$ can be used for node classification. For graph classification, they can be aggregated into a graph embedding $h_G$ with a 'readout' function,

$$h_G = \text{READOUT}(\{h_v^{(L)} \mid v \in V\}).$$

**Weisfeiler-Lehman (WL) Test.** The WL test [WL68] is a popular heuristic for the graph isomorphism problem. While the exact complexity of this problem remains unknown [Bab16], the WL test is a powerful heuristic capable of distinguishing a large family of graphs [BK79].

The WL test is as follows. Given a graph $G(V, E)$, initially all nodes are given the same fixed label $\ell_v^{(0)}$, say $\ell_v^{(0)} = 1$ for all $v \in V$. Then, in every iteration $k \geq 1$, each node $v$ is assigned the new label $\ell_v^{(k)} = \text{HASH}(\{\ell_u^{(k-1)} : u \in \mathcal{N}(v)\})$, where HASH is a 1-1 mapping (i.e., different multisets of labels are guaranteed to be hashed into distinct new labels). Performing this procedure on two graphs $G, G'$, the WL test declares them non-isomorphic if the label multiset $\{\ell_v^{(k)} : v \in V\}$ differs between the graphs at some iteration $k$. Note that the algorithm converges within at most $n$ iterations.

**Simulating WL with GNNs.** A GNN simulates WL deterministically if the node embeddings $h_v^{(k)}$ at each iteration $k$ constitute valid labels $\ell_v^{(k)}$ for the WL test. We also consider randomized simulation, where the GNN's weights are selected at random and we allow some small failure probability where distinct multisets of labels from iteration $k - 1$ are hashed into the same label at iteration $k$. This is captured by the next definition.

**Definition 1.1** (Successful Iteration of the WL Test). *A WL iteration gets existing labels $\{h_v : v \in V\}$ for all nodes, and outputs new labels $\{h'_v : v \in V\}$ given by*

$$h'_v = \phi\left(f\left(\{h_w : w \in \mathcal{N}(v)\}\right)\right),$$

*for an aggregate function $f$ and neural network $\phi$ with random weights. We say the iteration is **successful** if for all $v, u \in V$, the following holds:*

- *If $\{h_w : w \in \mathcal{N}(v)\} = \{h_w : w \in \mathcal{N}(u)\}$ then $h'_v = h'_u$ with probability 1, and*

- *If $\{h_w : w \in \mathcal{N}(v)\} \neq \{h_w : w \in \mathcal{N}(u)\}$ then $h'_v \neq h'_u$ with probability $1 - p$,*

*where the probability is over the choices of the random weights of $\phi$.*

To ensure the WL simulation is successful across $\text{poly}(n)$ iterations and for all pairs of nodes, we can set failure probability $p$ to $1/\text{poly}(n)$ and apply a union bound (see Prop. A.1 in Appendix A).

## 1.2 Overview of Our Techniques

In this section we give an overview of our GNN architectures for simulating WL. To explain our ideas in stages, we begin with a simpler construction of a polynomial size GNN. It is far larger than the ultimate polylogarithmic size we are aiming for, but forms a useful intermediate step toward our second and final construction.

**Construction 1 (Section 2).** Recall that the $k$th WL iteration, for a node $v$, aggregates the labels of its neighbors from the previous iteration, $\mathcal{H}_v^{(k-1)} := \{h_w^{(k-1)} : w \in \mathcal{N}(v)\}$, and hashes them into a new label $h_v^{(k)}$ for $v$. Our GNNs aggregate by summing, i.e., they sum $\mathcal{H}_v^{(k-1)}$ into $\mathcal{S}_v^{(k)} := \sum_{w \in \mathcal{N}(v)} h_w^{(k-1)}$, and then hash the sum into $h_v^{(k)}$ using a ReLU neural network of our choice.

If two nodes $u, v$ satisfy $\mathcal{H}_u^{(k-1)} \neq \mathcal{H}_v^{(k-1)}$, then WL assigns them distinct labels in iteration $k$, and therefore we want our construction to satisfy $h_u^{(k)} \neq h_v^{(k)}$ with probability at least $1 - p$ (as per Definition 1.1). This can fail in two places: either due to summing (if $\mathcal{S}_u^{(k)} = \mathcal{S}_v^{(k)}$ even though $\mathcal{H}_u^{(k-1)} \neq \mathcal{H}_v^{(k-1)}$) or due to hashing (if $h_u^{(k)} = h_v^{(k)}$ even though $\mathcal{S}_v^{(k)} \neq \mathcal{S}_v^{(k)}$). To avoid the first failure mode (summing), we use one-hot encodings for the node labels, meaning that for every node $v$ and iteration $k$, $h_v^{(k)}$ is a one-hot vector in $\{0,1\}^F$ (for $F$ to be determined shortly). This ensures that if $\mathcal{H}_u^{(k-1)} \neq \mathcal{H}_v^{(k-1)}$ then $\mathcal{S}_u^{(k)} \neq \mathcal{S}_v^{(k)}$ due to the linear independence of the one-hot vectors. To handle the second failure mode (hashing), we use random hashing into $F = O(1/p)$ buckets, rendering the collision probability no more than $p$. That is, we let $h_v^{(k)} = \text{One-Hot}(g(\mathcal{S}_v^{(k)}))$ where $g : \mathbb{Z}^F \to \{1, \ldots, F\}$ is chosen at random from an (approximately) universal hash family. We use the simple hash function $g(x) = \langle a, x \rangle \mod F$ where $a$ is random vector. It can be implemented with a ReLU neural network, since the dot product is just a linear layer, and the mod operation can be implemented with constant width and logarithmic depth, by a reduction to the "triangular wave" construction due to Telgarsky [Tel16] (see Appendix B.1). Finally, we show that turning the index of the hash bucket into a one-hot vector can be done with $O(F)$ ReLU units and constant depth.

Since our desired collision probability is $p = 1/\text{poly}(n)$, we set $F = O(1/p) = \text{poly}(n)$. Since our one-hot vectors are of dimension $F$, the resulting GNN has width $F = \text{poly}(n)$ (and depth $O(\log n)$ due to implementing the mod operation with ReLUs), and thus total size $\text{poly}(n)$.

**Construction 2 (Section 3).** We now proceed to describe our polylogarithmic size construction. The weak point in the previous construction was the wasteful use of one-hot encoding vectors, which caused the width to be $F$. In the current construction, we still wish to hash into $F$ bins — that is, to have $F$ distinct possible labels in each iteration — but we aim to represent them using bitstrings of length $O(\log F)$, thus exponentially improving the width of the network. That is, for every node $v$ and iteration $k$, the label $h_v^{(k-1)}$ would now be a vector in $\{0,1\}^{O(\log F)}$. The challenge is again to avoid the two failure modes above, ensuring that the failure probability does not exceed $p$. Since we cannot use one-hot encoding, we need to devise another method to avoid the first failure mode, i.e., ensure that with high probability $\mathcal{S}_u^{(k)} \neq \mathcal{S}_v^{(k)}$ if $\mathcal{H}_u^{(k-1)} \neq \mathcal{H}_v^{(k-1)}$.

To this end, suppose for a moment that we had access to a truly random vector $a \in \{0,1\}^F$. Then each node $v$, instead of sending the one-hot encoding of its label $h_v^{(k-1)}$, we could instead send

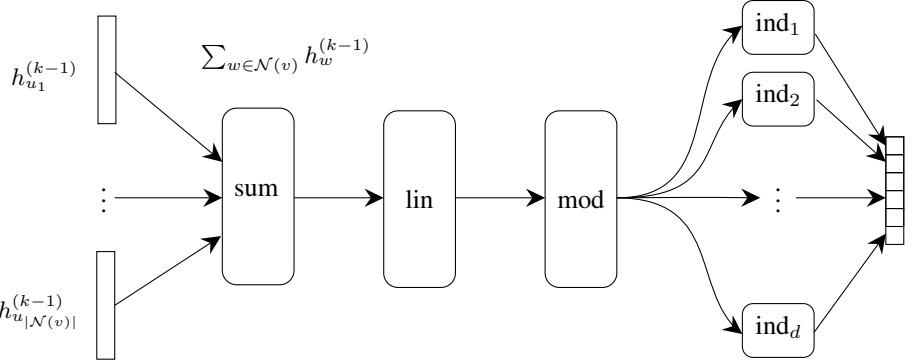

Figure 1: Constructions of our GNNs. Construction 1 (section 2): The layer "sum" sums the input vectors $\{h_w^{(k-1)} : w \in \mathcal{N}(v)\}$. "lin" is a linear layer that computes the dot product with a random vector and outputs a single scalar. "mod" is a neural network that computes the input scalar modulo $F$, using Theorem 2.2. Each "$\text{ind}_i$" is a neural network that outputs 1 if its input equals $i$ and outputs 0 otherwise, using eq. (2.1). The $i$th coordinate of the output vector $h_v^{(k)}$ equals the output of $\text{ind}_i$. Construction 2 (section 3): Similar, except that now each $\text{ind}_i$ is a neural network that gets an input integer $j \in [F]$ and outputs the $j$th coordinate of a vector $a^i$ sampled from an $\epsilon$-biased space, using Theorem C.1. This reduces the total number of units of the network from $\text{poly}(n)$ to $\text{polylog}(n)$.

the dot product $\langle a, h_v^{(k-1)} \rangle$, which is the single bit $a_{h_v^{(k-1)}}$. Each node $u$ thus receives the bits $\left\{ a_{h_w^{(k-1)}} : w \in \mathcal{N}(w) \right\}$ from its neighbors and aggregates them into the sum $\sum_{w \in \mathcal{N}(u)} \langle a, h_w^{(k-1)} \rangle$, which, by linearity, is equal to $\langle a, \mathcal{S}_u^{(k)} \rangle$ (using the notation from Construction 1). It is easy to observe that if $\mathcal{S}_u^{(k)} \neq \mathcal{S}_v^{(k)}$ then $\langle a, \mathcal{S}_u^{(k)} \rangle \neq \langle a, \mathcal{S}_v^{(k)} \rangle$ with probability at least 0.5. Repeating this process $\log F$ independent times decreases the collision probability to the requisite $1/F$. Thus, we can define a new labeling scheme $\bar{h}_v^{(k)} \in \{0,1\}^{\log F}$ that concatenates $\log F$ dot products with independent random vectors $a^1 \ldots a^{\log F} \in \{0,1\}^F$ as just described, failing at the summing operation with probability at most $1/F$. The second failure mode (hashing) can again be handled as before.

That catch is that, since $a$ has length $F$, the overall number of parameters in the GNN would again be at least $F$. To avoid this, we appeal to the computational theory of pseudorandomness. The idea is to replace the random vector $a$ with an efficient pseudorandom analog. Note that the above approach goes through even if the probability that $\langle a, \mathcal{S}_u^{(k)} \rangle \neq \langle a, \mathcal{S}_v^{(k)} \rangle$ is slightly larger than 0.5, say $0.5 + \epsilon$ for a small constant $\epsilon$. It is well-known in complexity theory that there exist pseudo-random generators, called $\epsilon$-*biased spaces*, that generate vectors $a$ satisfying this property given only $O(\log F + \log(1/\epsilon))$ truly random bits.[3] Crucially, each bit of $a$ can be computed using a threshold circuit with size polynomial in $O(\log F + \log(1/\epsilon))$ and constant depth (Theorem C.1), which translates to a ReLU neural network with the same parameters (Lemma C.2). Using these generators in our GNN to implement a pseudorandom ($\epsilon$-biased) analog of $a$ yields our final construction.

## 2 First Construction: Polynomial-size GNN

Our first construction towards Definition 1.1 is exponentially larger compared to our final optimized construction of Section 3. Nevertheless, it is instructive and motivates our optimized construction.

Let $h_u^{(k)}$ denote the label of a vertex $u$ in the $k$th iteration. For our first construction, we will always maintain the invariant that $h_u^{(k)}$ will be a one-hot encoded vector in $\{0,1\}^F$ for all $u \in V$ and all iterations $k$. $F > 2n$ will be a prime which also satisfies $F = O(\text{poly}(n))$. As stated previously, the aggregate function $f$ will just be the sum function. Our construction for the neural network used in the $k$th iteration, $\phi^{(k)}$, will take in the sum of the neighbors labels according to Equation (1.1) and output a one-hot encoded vector in $\{0,1\}^F$.

---

[3]Technically, they guarantee this property only when $\mathcal{S}_u^{(k)}$ are binary vectors, but Lemma 3.2 shows how to extend this property to general integer vectors as well.

**Implementation of Neural Network.** Our construction for $\phi^{(k)}$ is the following: First recall the notation from (simplified) Equation 1.1:

$$h_v^{(k)} = \phi^{(k)}\left(f\left(\left\{h_w^{(k-1)} : w \in \mathcal{N}(v)\right\}\right)\right).$$

1. For every node $v$, the input to $\phi^{(k)}$ is the sum of feature vectors of neighbors from the prior iteration, $\sum_{w \in \mathcal{N}(v)} h_w^{(k-1)}$, which is returned by the summing function $f$. Given any such input $x \in \mathbb{Z}^F$, $\phi^{(k)}$ computes the inner product of $x$ with a vector $a \in \mathbb{Z}^F$ where each entry of $a$ is a uniformly random integer in $[F] := \{0, \ldots, F-1\}$.

2. $\phi^{(k)}$ then computes $\langle x, a \rangle \bmod F$.

3. Finally, we represent the value of $z = \langle x, a \rangle$ as a one-hot encoded vector in $\mathbb{Z}^F$ where the $z$-th entry is equal to 1 and all other entries are 0's.

Altogether, $\phi^{(k)}$ can be summarized as: $h_v^{(k)} = \text{One-Hot}\left(\left\langle \sum_{w \in \mathcal{N}(v)} h_w^{(k-1)}, a \right\rangle \bmod F\right)$.

Note that we set the initial labels $h_u^{(0)}$ to be the same starting vector for all vertices (any one-hot vector). This matches the WL test which also initializes all nodes with the same initial label. Furthermore, the weights of $\phi^{(k)}$ are independent: the random vector $a$ is sampled independently for each iteration.

**Correctness of Construction.** The following lemma proves that the above construction satisfies the requirement of Definition 1.1. Its proof is given in Appendix B.

**Lemma 2.1.** *Let $\{h_w^{(k-1)} : w \in \mathcal{N}(v)\}$ and $\{h_w^{(k-1)} : w \in \mathcal{N}(u)\}$ denote the multiset of neighborhood labels for vertices $v$ and $u$ respectively. If the multisets are distinct then the labels computed for $v$ and $u$ in the $k$th iteration are the same with probability at most $O(1/F)$. If the multisets are the same then the labels are the same, i.e., the $k$th iteration is successful according to Definition 1.1.*

**Complexity of the GNN.** We now evaluate the size complexity of implementing our construction via a neural network $\phi$. Note that Step 1 of the construction can be done with 1 layer as it simply involves taking an inner product. The main challenge is to implement the modulo function. We give the following construction in Section B.1 of the appendix.

**Theorem 2.2.** *Suppose $F = \text{poly}(n)$. There exists an explicit construction of a network which computes modulo $F$ in the domain $\{0, \ldots, nF\}$ using a ReLU network with $O(\log n)$ hidden units and $O(\log n)$ depth. More generally, given an integer parameter $t > 0$, the function can be computed with $O((Fn)^{O(1/t)} \log n)$ hidden units and $O(t)$ depth.*

Directly appealing to the theorem above, we can implement modulo $F$ required in Step 2 of the construction using a neural network with $O(\log n)$ units, depth $O(\log n)$. In addition, we need only $O(\log n)$ bits to represent the weights.

Finally, Step 3 of our construction requires outputting a one hot encoding. We can do this by inputting $z$ (the output of Step 2 of the construction) into $F$ indicator functions, each of which detect if $z$ is equal to a particular integer in $[F]$. Each indicator function can be implemented via $O(1)$ ReLU nodes as follows. Let

$$g(x) = \text{ReLU}(\min(-\text{ReLU}(2x - 1/2) + 1, \text{ReLU}(2x + 1))) \tag{2.1}$$

which can be easily implemented as a ReLU network. (Note $\min(a, b) = a + b - \max(a, b)$ and $\max(a, b) = \max(a - b, 0) + b = \text{ReLU}(a - b) + b$.) It can be checked that $g(0) = 1$ and $g(x) = 0$ for all other integers $x \neq 0$ and that $g$ can be implemented with $O(1)$ ReLU function compositions. Thus, Step 3 of the construction requires $O(1)$ hidden layers and $O(F)$ total hidden units. Altogether we have proven the following result.

**Theorem 2.3.** *There exists a construction of a neural network $\phi$ which performs a successful iteration according to Definition 1.1 with failure probability $p = O(1/|F|)$. $\phi$ has depth $O(\log n)$, $O(F)$ hidden units, and requires $O(\log(nF))$ bits of precision. Furthermore, all labels in all iterations are vectors in $\{0, 1\}^F$. More generally, given an integer parameter $t > 0$, the function can be computed with $n^{O(1/t)} \text{polylog}(n)$ hidden units and $O(t)$ depth.*

**Remark 2.4.** *In the standard WL test, the number of iterations is chosen to be $O(n)$. Thus the right setting of $F$ in Theorem 2.3 is $F = O(\text{poly}(n))$ which gives us depth $O(\log n)$, $O(\text{poly}(n))$ hidden units, and requires $O(\log n)$ bits of precision in addition to labels in dimension $O(\text{poly}(n))$.*

## 3 Second Construction: Polylogarithmic-size GNN via Pseudo-randomness

We now present a more efficient construction of a GNN which simulates the WL test with an exponential improvement in the number of hidden units and label size. To motivate the improvement, we consider Step 3 of the prior construction which outputs a one-hot encoding. The one-hot encoding was useful as it allowed us to index into a uniformly random vector $a$ (which we then sum over mod $F$ in order to hash the neighborhood's labels). However, this limited us to use feature vectors of a large dimension and required many hidden units to create one-hot vectors. Instead of working with one-hot encodings as an intermediary, we will directly compute the entries of the random vector $a$ as needed. This has two advantages: we can significantly reduce the dimension of the feature vectors as well as reduce the total size of the neural networks used. We accomplish this via using *pseudo-random* vectors whose entries can be generated as needed with a small ReLU neural network (see Corollary 3.3). This allows us to use node labels in dimension $O(\log n)$ as opposed to $O(n)$.

The random vectors we employ have their entries generated from an $\varepsilon$-biased sample space. These are random vectors which are approximately uniform and they have been well-studied in the complexity-theory literature. We recall some definitions below.

**Definition 3.1** (Bias). *Let $X$ be a probability distribution over $\{0,1\}^m$. The bias of $X$ with respect to a set of indices $I \subseteq \{1,\ldots,m\}$ is defined as $\text{bias}_I(X) = \left| \mathbb{P}_{x \sim X}\left[\sum_{i \in I} x_i = 0\right] - \mathbb{P}_{x \sim X}\left[\sum_{i \in I} x_i = 1\right] \right|$ where each sum is taken modulo 2 and the empty sum is defined to be 0.*

**Definition 3.2** ($\varepsilon$-biased Sample Space). *A probability distribution over $\{0,1\}^m$ is called an $\varepsilon$-biased sample space if $\text{bias}_I(X) \leq \varepsilon$ holds for all non-empty subsets $I \subseteq \{1,\ldots,m\}$.*

Note that the uniform distribution has bias 0. We now state our construction for the neural network used in the $k$th iteration, $\phi^{(k)}$. We recall that $h_u^{(k)}$ denotes the label of a vertex $u$ in the $k$th iteration.

**Implementation of Neural Network.** Our construction for $\phi^{(k)}$ is the following:

1. The input vectors of $\phi^{(k)}$, which are of the form $h_v^{(k-1)}$ for the $k$th iteration, are each assumed to be a feature vector in $\mathbb{Z}^{C \log n}$ for a sufficiently large constant $C$. The output of $\phi^{(k)}$ will also be a feature vector in $\mathbb{Z}^{C \log n}$. Our aggregation function $f$ will again be the summation function.

2. Let $F$ be a prime of size $\text{poly}(n)$ which is at least $2n$.

3. For each node $v$, $\phi^{(k)}$ computes $z_v = \langle b, \sum_{w \in \mathcal{N}(v)} h_w^{(k-1)} \rangle \mod F$ where every entry of $b$ is uniformly random in $\{0,\ldots,F-1\}$. Note $\sum_{w \in \mathcal{N}(v)} h_w^{(k-1)}$ is the output of the aggregation $f$.

4. Let $a^t \in \{0,1\}^F$ for $t = 1,\ldots,C\log n$ be vectors which are independently drawn from an $\varepsilon$-biased sample space for a sufficiently small constant $\varepsilon$.

5. The output $h_v^{(k)}$ will be a $C \log n$ dimensional binary vector where the $t$-th coordinate is equal to the $z_v$-th coordinate of the vector $a^t$. In other words, $h_v^{(k)} = (a^t(z_v))_{t=1}^{C \log n}$ where $a^t(z_v)$ denotes the $z_v$-th coordinate of $a^t$.

**Correctness of Construction.** We now prove the correctness of our construction. We will refer to $z_v$ computed in Step 3 of the construction as the *index* of $v$ for the $k$th iteration. To prove the correctness of the above construction, it suffices to prove the lemma below which shows our construction satisfies Definition 1.1.

**Lemma 3.1.** *Let $\{h_w^{(k-1)} : w \in \mathcal{N}(v)\}$ and $\{h_w^{(k-1)} : w \in \mathcal{N}(u)\}$ denote the multiset of neighborhood labels for vertices $v$ and $u$ respectively. If the multisets are distinct then the labels computed for $v$ and $u$ in the $k$th iteration are distinct with probability $1 - 1/\text{poly}(n)$. If the multisets are the same then the labels are the same, i.e., the $k$th iteration is successful according to Definition 1.1.*

We first need the following auxiliary lemma about $\varepsilon$-biased sample spaces, proven in Section C.

**Lemma 3.2.** *Let $\mathcal{D}$ be a probability distribution over $\{0,1\}^m$ that is an $\varepsilon$-biased sample space. Then, for any $x, y \in \mathbb{Z}^m$ such that $x \neq y$, $\mathbb{P}_{a \sim \mathcal{D}}[\langle a, x \rangle = \langle a, y \rangle] \leq \frac{1}{2} + \frac{\varepsilon}{2}$.*

Note that this lemma is necessary, as we will be computing dot products of $a$ with integer vectors (over integers), not with binary vectors modulo 2.

We are now ready to prove Lemma 3.1.

*Proof of Lemma 3.1.* Let $x' = \sum_{w \in \mathcal{N}(v)} h_w^{(k-1)}$ denote the input for $v$ and analogously, define $y' = \sum_{w \in \mathcal{N}(u)} h_w^{(k-1)}$ to be the input for $u$. We first show that if $\{h_w^{(k-1)} : w \in \mathcal{N}(v)\}$ is not equal to (as multisets) $\{h_w^{(k-1)} : w \in \mathcal{N}(u)\}$ then $x' \neq y'$ with sufficiently large probability. We further consider the case that $k \geq 2$ since for $k = 1$ (the first iteration), the statement follows since all node labels are initialized to be the same. Let $z_v'$ be the indices computed in Step 3 of iteration $k - 1$ (which are used to construct the node labels $h_v^{(k-1)}$ in iteration $k - 1$). Note that there is a one to one mapping between $z_v'$ and $h_v^{(k-1)}$. Thus we can assume $\{z_w' : w \in \mathcal{N}(v)\} \neq \{z_w' : w \in \mathcal{N}(u)\}$ without loss of generality.

Let $\tilde{a}^1$ be the first $\varepsilon$-biased vector used in the previous neural network $\phi^{(k-1)}$. Note that the first entry of $x'$ is equal to $\sum_{w \in \mathcal{N}(v)} \tilde{a}^1(z_w)$, i.e., it is the dot product of $\tilde{a}^1$ with a suitable vector $x$ which is the sum of one-hot encoding of the neighborhood of $v$. The same statement is true for $y'$: the first entry of $y'$ is equal to the dot product of $\tilde{a}^1$ with a vector $y$ which represents the one-hot encoding of the neighborhood of $u$. This is because we computed the index $z_v'$ of every node in the previous iteration, as defined in Step 3 of the construction, and passed along the coordinate of $\tilde{a}^1$ which corresponds to this computed index. By assumption, we know that $x \neq y$. Therefore by Lemma 3.2, we have that $\mathbb{P}_{\tilde{a}^1}[(x')_1 = (y')_1] = \mathbb{P}_{\tilde{a}^1}(\langle \tilde{a}^1, x \rangle = \langle \tilde{a}^1, y \rangle) \leq 2/3$ for a suitable $\varepsilon$. By independence of vectors $\tilde{a}^t$, it follows that $\mathbb{P}[x' = y'] \leq (2/3)^{C \log n} \leq 1/\operatorname{poly}(n)$ for a suitable constant $C > 0$.

We now condition on $x' \neq y'$. Without loss of generality, suppose that their first coordinates, $x_1'$ and $y_1'$, differ. We know $x_1' \neq y_1' \bmod F$ since $x_1' \neq y_1'$, they are both non-negative and bounded by $n$, and $|x_1' - y_1'| \leq O(n)$ whereas $F$ is a prime at least $2n$. It follows that the probability of the event $\langle (x' - y'), b \rangle = 0 \bmod F$ is at most $1/F$. To see this, condition on all the entries of $b$ except $b_1$. Then $(x' - y')_1 \cdot b_1$ must be equal to a specific value modulo $F$ for $\langle (x' - y'), b \rangle = 0 \bmod F$ to hold, as desired. We now condition on this event which equivalently means we condition on $z_v \neq z_u$ (see Step 3 of the construction).

Now our task is to show that $h_v^{(k)} \neq h_u^{(k)}$ with sufficiently high probability. The first coordinate of $h_v^{(k)}$ is equal to $\langle a^1, e_{z_v} \rangle$ where $a^1$ is the first $\varepsilon$-biased vector considered in Step 4 of the construction and $e_{z_v}$ is the basis vector in dimension $C \log n$ which has a 1 entry only in the $z_v$ coordinate and 0 otherwise. Therefore by Lemma 3.2, we have that , $\mathbb{P}_{a^1}[(h_v^{(k)})_1 = (h_u^{(k)})_1] = \mathbb{P}_{a^1}[\langle a^1, e_{z_v} \rangle = \langle a^1, e_{z_u} \rangle] \leq 2/3$ for a suitable choice of $\varepsilon$ since $e_{z_u}$ and $e_{z_v}$ are distinct. By independence of vectors $a^t$, it follows that $\mathbb{P}[h_v^{(k)} = h_u^{(k)}] \leq (2/3)^{C \log n} \leq 1/\operatorname{poly}(n)$ for a suitable constant $C > 0$. This exactly means that the node labels of $u$ and $v$ in the next iteration are different with probability at least $1 - 1/\operatorname{poly}(n)$, as desired. Lastly, it is clear that if the multiset of neighborhood labels of $u$ and $v$ are the same, then $x' = y'$ and it's always the case that $h_v^{(k)} = h_u^{(k)}$. $\qquad\square$

**Complexity of the GNN.** We now analyze the overall complexity of representing $\phi^{(k)}$ as a ReLU neural network. First we state guarantees on generating $\varepsilon$-based vectors using a ReLU network. The following corollary is proven in Appendix C.

**Corollary 3.3.** *Let $s = O(\log F + \log(1/\varepsilon))$. For every $\varepsilon$ and $F$, there exists an explicit ReLU network $C : \{0,1\}^s \cup [F] \to \{0,1\}^F$ which takes as input $s$ uniform random bits and an index $i \in F$ and outputs the ith coordinate of an $\varepsilon$-biased vector in $\{0,1\}^F$. $C$ uses $O(\log F)$ bits of precision and has $\operatorname{poly}(s)$ hidden units. More generally, given an integer parameter $t > 0$, the function can be computed with $n^{O(1/t)} \operatorname{polylog}(n)$ hidden units and $O(t)$ depth.*

We can now analyze the complexity of our construction. The complexity can be computed by analyzing each step of the construction separately as follows:

1. For every node $v$, the sum of feature vectors of neighbors from the prior iteration, $\sum_{w \in \mathcal{N}(v)} h_w^{(k-1)}$, is returned by the aggregate function $f$.

2. The inner product with the random vector $b$ in Step 3 of the construction can be computed using one layer of the network. Then computing modulo $F$ can be constructed via Theorem 2.2.

3. Given the inner product value $z_v$ which is the output of Step 3 of the construction, we compute all of the $O(\log n)$ coordinates of $h_v^{(k)}$ in parallel. We recall that each coordinate of $h_v^{(k)}$ is indexing onto $O(\log n)$ $\varepsilon$-biased random vectors and we use the *same* index for all vectors, namely the $z_v$-th index. This can be done as follows. We first have $O(\log n)$ edges fanning-out from the node which computes $z_v$. For all $t = 1, \ldots, O(\log n)$, the other endpoint of the $t$-th fan-out edge computes the value $a^t(z_v)$ where $a^t$ is the $t$-th $\varepsilon$-biased vector as stated in Steps 4 and 5 of the construction. This can be done by appealing to the construction guaranteed by Corollary 3.3. The result of this computation is exactly $h_v^{(k)}$.

Altogether, we have proven the following theorem.

**Theorem 3.4.** *There exists a construction of $\phi$ which performs a successful WL iteration according to Definition 1.1 with $p \leq 1/\operatorname{poly}(n)$. $\phi$ has depth $O(\log n)$, $O(\operatorname{poly}(\log n))$ hidden units, and requires $O(\log n)$ bits of precision. All labels in all iterations are binary vectors in $\{0, 1\}^{O(\log n)}$. More generally, given an integer parameter $t > 0$, the function can be computed with $n^{O(1/t)} \operatorname{polylog}(n)$ hidden units and $O(t)$ depth.*

# 4 Lower Bounds

We complement our construction with lower bounds on the label size and number of ReLU units required to simulate the WL test. We outline these two lower bounds below and defer the full details to Appendix D.

**Message Size.** Recall that in our construction, the message (label) size was $O(\log n)$ bits. Via communication complexity, we give a corresponding lower bound. In particular, we construct a graph on which any (randomized) communication protocol which simulates WL as in Definition 1.1 must send at least $\Omega(\log n)$ bits along one edge of the graph. As message-passing GNNs are a specific class of communication protocols, this immediately implies that the message sizes must have $\Omega(\log n)$ bits, so our construction is optimal in that respect.

The hard instance is formed by a graph which is a collection of disjoint star subgraphs of sizes ranging from 2 to $\Theta(\sqrt{n})$. In order to perform a valid WL coloring, each node must essentially learn the size of its subgraph, requiring $\Omega(\log(\sqrt{n})) = \Omega(\log n)$ bits of communication. In addition, this must be done in only 2 iterations as the depth of each subgraph is 2, so some node must send $\Omega(\log n)$ bits to its neighbors in a single round. See Appendix D.1 for the full details and proof.

**Number of ReLU Units.** In order to show a lower bound on the number of units needed to implement a successful WL iteration, we rely on prior work lower bounding the number of linear regions induced by a ReLU network (for instance [MPCB14]). In particular, these works show that ReLU networks induce a partition of the input space into $N$ convex regions (where $N$ is a function of the size of the network) such that the network acts as a linear function restricted to any given region. Using these results, we describe a fixed graph and a distribution over inputs to the neural network $S_u^{(k-1)}$ for all $u \in V$ (sums of the labels from the previous round) which includes $O(F)$ potential special pairs of nodes (where $F$ is defined such that inputs $S_u^{(k-1)} \in [F]^t$ for some $t$). For each such pair $u, v$, their neighborhoods $N(u), N(v)$ have different multisets of inputs, but both multisets of inputs sum to the same value. We show that if the number of linear regions is small, $N = o(F)$, then it is relatively likely that $u, v$ will be in the same linear region and thus their sums will collide: $S_u^{(k)} = S_v^{(k)}$ even while their neighborhoods had distinct inputs in the $(k-1)$st round.

This immediately gives a $\Omega(\log F)$ lower bound on the number of ReLU units (and thus number of parameters) with more refined depth/width tradeoffs given in Appendix D.2.1. Note that $F$ is the size of each coordinate in the *sum* of labels. Even if the labels are binary, $F$ can be as large as $n$, depending on the max degree in the graph, which implies a $\Omega(\log n)$ lower bound on the number of ReLU units. See Appendix D.2 for full details and proof.

# 5 Experiments

To demonstrate the expressivity of our construction, i.e., that our small-sized GNN reliably simulates the WL test, we perform experiments on both synthetic and real world data sets. Common to all of our experiments is that we start with some graph $G = (V, E)$ (either real world or generated with respect to some probability distribution). We then simulate a perfect run of the WL test on $G$ where any two nodes which receive different multisets of labels in iteration $k - 1$ get distinct labels in iteration $k$ with probability 1 as well as a run of our construction from[4] Section 3. At any point in time, the node labels induce partitions of $V$ where two nodes are in the same class if they have the same labels. Denote the partitions after $k$-iterations using the perfect simulation and our construction respectively by $\mathcal{P}_k$ and $\mathcal{P}'_k$. Letting $k_0$ be minimal such that $\mathcal{P}_{k_0-1} = \mathcal{P}_{k_0}$ (at which point the WL labels have converged), we consider the implementation using our GNN successful if $\mathcal{P}_k = \mathcal{P}'_k$ for all $k \leq k_0$, i.e., if the the simulation using our implementation induced the same partitions as a perfect runs. For all of our experiments it turned out that $k_0 \leq 5$ (see [BK22] for a discussion of this fast convergence).

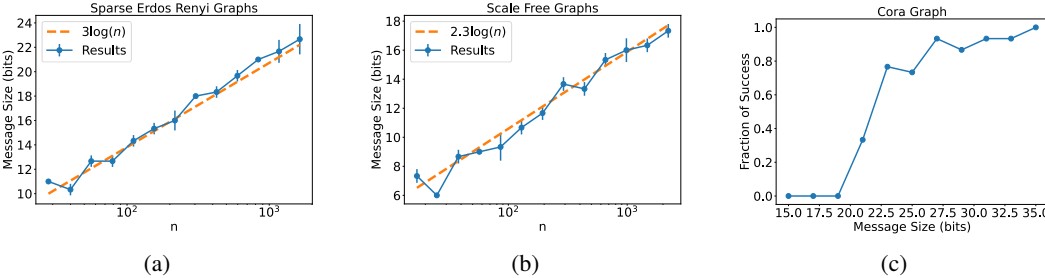

(a)  (b)  (c)

Figure 2: Plots of our experimental results on Erdős-Rényi graphs, scale free graphs, and the real world Cora graph. The vertical blue lines in (a) and (b) are the empirical standard deviations over the 5 independent samples of random graphs. Note that the $x$-axes in (a) and (b) are logarithmic.

**Sparse Erdős-Rényi Graphs**   We generated Erdős-Rényi random graphs $G(n, p)$ with $p = 20/n$ for a varying number of vertices $n$. For each value of $n$, we generated five such graphs and for each of these five graphs, we ran 10 independent trials of our GNN implementation with message sizes $t = 1, 2, \ldots$. Averaging over the five graphs, we report the minimal $t$ such that at least 70% of the 10 iterations successfully simulated the WL test. See Figure 2a. The average message size needed to achieve this is approximately $3 \log n$ where the logarithmic dependence on $n$ is as predicted theoretically and significantly improves on the linear message size required for prior constructions.

**Scale Free Graphs**   We generated samples of the scale free graphs from [BBCR03] with a varying number of vertices $n$ using the implementation from [HSS08]. Our experiment design was the same as for Erdős-Rényi random graphs. See Figure 2b.

**Cora Graph**   We finally ran experiments on the real world graph Cora[5] which is the citation network of $n = 2708$ scientific publications. We simulated our GNN with varying message lengths, for each message length reporting the fraction of successful runs of 30 independent trials. See Figure 2c for a plot of the results. We see that with message length 35, all of the 30 trials successfully simulated the WL test.

**Acknowledgements**   Anders Aamand is supported by DFF-International Postdoc Grant 0164-00022B from the Independent Research Fund Denmark. This research was also supported by the NSF TRIPODS program (award DMS-2022448), NSF award CCF-2006664, Simons Investigator Award, MIT-IBM Watson AI Lab, GIST- MIT Research Collaboration grant, NSF Graduate Research Fellowship under Grant No. 1745302, and MathWorks Engineering Fellowship.

---

[4]Since our goal is to test whether our protocol correctly simulates WL test with small messages, we are not implementing the actual GNNs but instead we are simulating their computation. Further, for simplicity, we replaced the $\varepsilon$-biased sample space with a random string, which guarantees $\varepsilon = 0$.

[5]https://graphsandnetworks.com/the-cora-dataset/

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
