# A   Omitted Proofs of Section 1

**Proposition A.1.** *Let $\ell_v^{(T)}$ be the labels of nodes after running the WL test for $T$ iterations on an input graph $G$ with initial labels $\ell_v^{(0)}$ all equal and suppose $p \leq \delta/(n^2 T)$. Suppose $T$ independent GNN iterations are successful according to Definition 1.1 and the output labels of iteration $i$ are the input labels of iteration $i + 1$ for all $i$ with the initial labels being equal to $\ell_v^{(0)}$. Let $h_v^{(T)}$ denote the node labels which are the output of the final iteration of the GNN. Then for all $v_1, v_2 \in V$, the following statements hold with probability $1 - \delta$:*

- *If $\ell_v^{(T)} = \ell_u^{(T)}$ then $h_v^{(T)} = h_u^{(T)}$ and*
- *If $\ell_v^{(T)} \neq \ell_u^{(T)}$ then $h_v^{(T)} \neq h_u^{(T)}$.*

*Proof.* Consider some iteration $i \leq T$. Suppose we have the following guarantee on the node label inputs for the $i$th iteration (note the inputs are the output labels of the previous iteration):

- If $\ell_v^{(i-1)} = \ell_u^{(i-1)}$ then $h_v^{(i-1)} = h_u^{(i-1)}$ and
- If $\ell_v^{(i-1)} \neq \ell_u^{(i-1)}$ then $h_v^{(i-1)} \neq h_u^{(i-1)}$.

Given the guarantee for iteration $i-1$, we show that the same guarantee is true for the labels outputted in the $i$th iteration with probability $1 - p$. Consider what the WL test does given input labels $\ell_{v_1}^{(i-1)}$: it assigns the same node labels to all pairs of vertices with the same multiset of neighborhood (input) labels and different labels for all pairs of vertices with different multiset of neighborhood (input) labels. This is exactly the same guarantee as Definition 1.1, except we also have to union bound over all $\leq n^2$ pairs of nodes in Definition 1.1. Thus, the invariant stated in the beginning of the proof also holds for the $i$th iteration with probability $1 - n^2 p$. Now applying a union bound across all $T$ iterations, the total failure probability is $n^2 T p \leq \delta$, as desired. $\qquad\square$

# B   Omitted Proofs of Section 2

*Proof of Lemma 2.1.* Since all vectors $h_u^{(k)}$ are always one-hot encoded vectors, if two nodes have differing neighborhood multisets of labels, then the sum of the one-hot encoding vectors of each neighborhoods will also be different. Suppose we are in this case and let $y$ and $y'$ denote the sum of the neighborhood labels of $v$ and $u$ respectively. Without loss of generality, suppose $y$ and $y'$ differ in the first coordinate, $y_1$ and $y_1'$. Then $\langle a, y - y' \rangle \equiv 0 \bmod F$ if and only if $a_1(y_1 - y_1') \equiv c \bmod F$ where $c = \sum_{i=2}^F a_i(y_i' - y_i)$. Condition on the event that $a_1 \not\equiv 0 \bmod F$ which happens with probability $1 - 1/F$. We now claim that $y_1' - y_1 \not\equiv 0 \bmod F$. Indeed, $0 \leq y_1 \neq y_1' \leq n$ so $0 \neq |y_1 - y_1'| \leq n$. Since $F > 2n$, it cannot be the case that $y_1 - y_1' \equiv 0 \bmod F$ which means that the multiplicative inverse of $(y_1 - y_1') \bmod F$, denoted as $(y_1 - y_1')^{-1}$, is well-defined and unique. Thus for $a_1(y_1 - y_1') \equiv c \bmod F$, we must have $a_1 \equiv c(y_1 - y_1')^{-1} \bmod F$ which happens with probability $O(1/F)$. Altogether, we have that $\mathbb{P}_a[\langle a, y \rangle \equiv \langle a, y' \rangle \bmod F] \leq O(1/F)$, as desired. Conditioning on $\langle a, y \rangle \not\equiv \langle a, y' \rangle \bmod F$, we have that $h_v^{(k)} \neq h_u^{(k)}$ as their single non-zero coordinates are distinct. Finally, we can easily check that if the multiset of neighborhoods of $u$ and $v$ are the same, then $h_v^{(k)} = h_u^{(k)}$ always holds. $\qquad\square$

## B.1   Implementation of Modulo $F$ via ReLU Network

We now give an efficient ReLU network construction of the function computing modulo $F$, thereby proving Theorem 2.2. First we define the function $\mathrm{TW}_F : \{0, \ldots, NF\} \to \{0, \ldots, F-1\}$, also known as the "triangular wave" function:

$$\mathrm{TW}_F(x) = \mathrm{T}_F(x \bmod 2F)$$

where

$$\mathrm{T}_F(x) = \begin{cases} x, & \text{if } x \leq F \\ 2F - x, & \text{if } x > F. \end{cases}$$

A result of [Tel16] implements the function $\mathrm{TW}_F$ using low-depth ReLU networks.

**Theorem B.1** (Lemma 3.8 and Corollary 3.9 in [Tel16]). *Suppose $F = \mathrm{poly}(n)$. The function* $\mathrm{TW}_F : \{0, \ldots, nF\} \to \{0, \ldots, F - 1\}$ *can be implemented using a neural network with $O(\log n)$ hidden units and $O(\log n)$ depth.*

We now reduce the modulo $F$ case to the construction of the triangular wave function.

**Theorem 2.2.** *Suppose $F = \mathrm{poly}(n)$. There exists an explicit construction of a network which computes modulo $F$ in the domain $\{0, \ldots, nF\}$ using a ReLU network with $O(\log n)$ hidden units and $O(\log n)$ depth. More generally, given an integer parameter $t > 0$, the function can be computed with $O((Fn)^{O(1/t)} \log n)$ hidden units and $O(t)$ depth.*

*Proof.* Assume without loss of generality that $F$ is odd. Given an integer $z \in \mathbb{Z}$, we first compute $\mathrm{TW}_F(z)$ and $\mathrm{TW}_{F/2}(z)$. Note that even though $F/2$ is non-integral, we can still compute it. Looking for a general real number $z \in [0, 2F]$, we have that $\mathrm{TW}_F(z) = \mathrm{TW}_{F/2}(z)$ if and only if $z \in [0, F/2]$ or $z \in [3F/2, 2F]$. In addition, if $z$ is an integer in $[0, 2F]$ but not in $[0, F/2] \cup [3F/2, 2F]$, then $\mathrm{TW}_F(z) - \mathrm{TW}_{F/2}(z) \geq 1$. Noting that $\mathrm{ReLU}(1 - \mathrm{ReLU}(1 - x)) = 0$ for $x = 0$ and $\mathrm{ReLU}(1 - \mathrm{ReLU}(1 - x)) = 1$ for any $x \geq 1$, we therefore have that

$$\mathrm{ReLU}(1 - \mathrm{ReLU}(1 - \mathrm{TW}_F(z) + \mathrm{TW}_{F/2}(z)))$$

equals 0 if $z \in \mathbb{Z}$ and $0 \leq (z \bmod 2F) \leq (F - 1)/2$ or $(3F + 1)/2 \leq (z \bmod 2F) \leq 2F - 1$, and equals 1 if $z \in \mathbb{Z}$ and $(F + 1)/2 \leq (z \bmod 2F) \leq (3F - 1)/2$. So, if we consider shifting $z$ by $(F - 1)/2$, we can make this equal 1 if and only if $z$ is between $F$ and $2F - 1 \bmod F$. Therefore, for integers $z$,

$$\mathrm{ReLU}\left(1 - \mathrm{ReLU}\left(1 - \mathrm{TW}_F\left(z - \frac{F-1}{2}\right) + \mathrm{TW}_{F/2}\left(z - \frac{F-1}{2}\right)\right)\right)$$
$$= \begin{cases} 0 & 0 \leq (z \bmod 2F) \leq F - 1 \\ 1 & F \leq (z \bmod 2F) \leq 2F - 1 \end{cases}.$$

Now, for simplicity, let us define the function above as $g(z)$. Note that when $g(z) = 0$, then $\mathrm{TW}_F(z) = z \bmod F$, and when $g(z) = 1$, then $\mathrm{TW}_F(z) = F - (z \bmod F)$, so $z \bmod F = F - \mathrm{TW}_F(z)$. Therefore, we have that for any integer $z$, $z \bmod F = |F \cdot g(z) - \mathrm{TW}_F(z)|$. But note that we can write

$$|x| = \max(x, -x) = \max(2x, 0) - x = \mathrm{ReLU}(2x) - x.$$

Therefore, we have that

$$z \bmod F = \mathrm{ReLU}(2F \cdot g(z) - 2 \cdot \mathrm{TW}_F(z)) - F \cdot g(z) + \mathrm{TW}_F(z). \qquad \square$$

## C  Omitted Proofs for Section 3

*Proof of Lemma 3.2.* Since $\langle a, x \rangle - \langle a, y \rangle = \langle a, x - y \rangle$, by writing $z = x - y \in \mathbb{Z}^m$, it suffices to show that if $z$ is nonzero, then $\mathbb{P}_{a \sim \mathcal{D}}[\langle a, z \rangle = 0] \leq \frac{1}{2} + \frac{\varepsilon}{2}$.

Let $k$ represent the largest nonnegative integer such that $2^k | z_i$ for all $i \in [m] := \{1, 2, \ldots, m\}$, and let $z' = z/2^k$. Then, by replacing $z$ with $z'$, we have that $\mathbb{P}_{a \sim \mathcal{D}}[\langle a, z \rangle = 0]$ if and only if $\mathbb{P}_{a \sim \mathcal{D}}[\langle a, z' \rangle = 0]$ (since we are just dividing by $2^k$) and $z_i'$ is odd for at least one value of $i \in [m]$. So, it suffices to show that $\mathbb{P}_{a \sim \mathcal{D}}[\langle a, z' \rangle = 0] \leq \frac{1}{2} + \frac{\varepsilon}{2}$. Note that by reducing modulo 2, it suffices to show that over $\mathbb{GF}_2$, $\mathbb{P}_{a \sim \mathcal{D}}[\langle a, z' \rangle \equiv 0 \bmod 2] \leq \frac{1}{2} + \frac{\varepsilon}{2}$, because then the probability over the integers is either the same or lower.

Let $I$ be the subset of $i \in [m]$ for which $z_i'$ is odd. Note that $I$ is nonempty since we know $z_i'$ is odd for at least one value of $i \in [m]$. So, by the definition of $\varepsilon$-biased sample spaces, we know that over $\mathbb{GF}_2$, $\left|2 \cdot \mathbb{P}_{a \in \mathcal{D}}\left[\sum_{i \in I} a_i = 0\right] - 1\right| \leq \varepsilon$, which means that $\mathbb{P}_{a \in \mathcal{D}}\left[\sum_{i \in I} a_i = 0\right] \leq \frac{1}{2} + \frac{\varepsilon}{2}$. But indeed $\sum_{i \in I} a_i \equiv \langle a, z' \rangle$ since $i \in I$ precisely when $z_i'$ is odd, so this completes the proof. $\qquad \square$

## C.1 Proof of Corollary 3.3

We first need to define the circuit class $\text{TC}^0$.

**Definition C.1** (Threshold Gate). *For inputs $x_1, \ldots, x_m \in \{0, 1\}$ the output of a threshold gate,* TH, *is*

$$\text{TH}(x_1, \ldots, x_m) = \begin{cases} 1 & \sum_{i=1}^m a_i x_i \geq \theta \\ 0 & otherwise \end{cases}$$

*where $\theta, a_1, \ldots, a_m \in \mathbb{Z}$. $\theta, a_1, \ldots, a_m$ may depend on $n$ but they do not depend on the input $x_1, \ldots, x_m$.*

**Definition C.2** ($\text{TC}^0$ Circuit Class). $\text{TC}^0$ *is the class of boolean functions computed by constant-depth $\text{poly}(m)$-size circuits with threshold gates.*

It is known that $\varepsilon$-biased vectors can be generated using an efficient circuit in $\text{TC}^0$.

**Theorem C.1** (Theorem 14 in [HV06], Restated). *Let $s = O(\log F + \log(1/\varepsilon))$. For every $\varepsilon$ and $F$, there exists an explicit $\text{TC}^0$ circuit $C : \{0,1\}^s \cup \{0,1\}^{\lceil log_2 F \rceil} \to \{0,1\}$ which takes as input $s$ uniform random bits and an index $i \in [F]$ and outputs the ith coordinate of an $\varepsilon$-biased vector in $\{0,1\}^F$. $C$ uses $\text{poly}(s)$ threshold gates.*

Note that the guarantees of Theorem C.1 are not directly applicable since we need to use a ReLU network instead of threshold gates. Nevertheless, since the circuit $C$ guaranteed by Theorem C.1 has integer inputs in all gates, we can easily approximate each threshold gates using an appropriately scaled ReLU. This is a straightforward and known reduction but we briefly outline a procedure in Lemma C.2.

**Lemma C.2.** *Consider the threshold gate* TH$: \{0,1\}^m \to \{0,1\}$ *which computes the threshold $\sum_{i=1}^m a_i x_i \geq \theta$. Assume that $a_i, \theta$ are all integers bounded by $\text{poly}(m)$.* TH *can be computed by a ReLU network using $O(\log m)$ bits of precision and a constant number of parameters.*

*Proof.* Consider the function

$$g(x) = \text{ReLU}(-\text{ReLU}(-x+2)).$$

It is 0 for all integers $x \leq 0$ and 1 for all integers $\geq 1$, i.e., it computes the threshold "$x \geq 0$". By shifting and scaling $g$, we can now compute the threshold "$x \geq \theta$" for any integer $\theta$. Finally, the sum $\sum_{i=1}^m a_i x_i$ can be computed using one additional layer. Since all parameters are integers, we only require $O(\log m)$ bits of precision to store the shifting and scaling factors. $\qquad\square$

Lastly, we remark that as per the definition of a threshold gate in Definition C.1, Theorem C.1 requires the index $i \in [F]$ to be inputted as a binary string with its bits given on individual nodes. However, this presents a slight inconsistency with the statement of Theorem C.1 and its corollary, Corollary 3.3 which is used in the construction of Section 3. Specifically, Step 3 of the construction of Section 3 outputs the actual integer $i \in [F]$ which we use as the index for our $\varepsilon$-biased vector, which does not match the format required by Theorem C.1. This inconsistency is straightforward to fix without having any impact whatsoever in the asymptotic size complexity of the neural network. We simply take the integer $i$ outputted by Step 3 of the construction and compute the $j$th bit of $i$ for all $1 \leq j \leq O(\log F)$ in parallel. The $j$th bit is exactly equal to 0 if and only if $(i \bmod 2^{j+1}) < 2^j$ and 1 otherwise. Note that $2^{j+1} = O(F)$ for all $j$ and we can easily compute each $\bmod 2^{j+1}$ by appealing to Theorem 2.2. This only requires $O(1)$ extra depth and an additional $O(\text{poly}(\log n))$ hidden units and $O(\log n)$ bits of precision. The more general trade-off of Theorem 3.4 also readily holds.

# D Lower bounds

In this appendix we provide lower bounds on the complexity of graph neural networks that are able to simulate the WL test. We present both a communication complexity lower bound and a lower bound on the number of ReLU units of the GNN. More concretely, in Section D.1, we prove that in order to maintain the invariant that with at least some constant probability, nodes with isomorphic neighborhoods get the same label while nodes with non-isomorphic neighborhoods get different

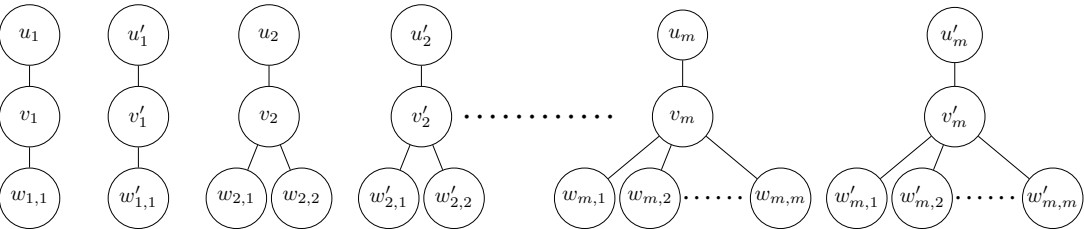

Figure 3: The graph $G$ used for Theorem D.1, showing the disconnected pieces $G_1, G'_1, G_2, G'_2, \ldots, G_m, G'_m$. Note that for all $k$, $G_k$ and $G'_k$ are isomorphic.

labels, some message sent between nodes must be of length at least $\Omega(\log n)$. This bound matches the upper bound of Theorem 3.4. Second, in Section D.2, we consider a more specific although still fairly general lower bound model which captures the implementation of the WL test using neural networks. We suppose that the messages sent between nodes are $t$-dimensional vectors with integral entries. We moreover suppose that each node combines its received messages by summing them to get a vector in $[F]^t$ (here, $[F] = \{0, 1, \ldots, F - 1\}$) and applying a collectively agreed upon neural network $\phi$ with at most $H$ ReLU units to this sum. We show that if the combination of summing neighborhoods and applying the neural network maps distinct multisets to distinct elements with at least some constant probability, then $H = \Omega(\log F)$. Moreover, parametrizing in terms of the depth and width of the neural network, we obtain a more fine-grained lower bound, demonstrating that for shallow neural networks, we need even more ReLU units. In Remark D.5, we point out that our lower bound holds even if the aggregation function $f$ is itself a neural network with a bounded number of ReLU units. As a node in an $n$-node graph could have up to $n - 1$ neighbours, we need at least $F = \Omega(n)$ in order to store the sum of the messages from the neighbors of the nodes. With this assumption, the lower bound thus becomes $\Omega(\log n)$ which matches our upper bound up to $\mathrm{polylog}(n)$ factors. It remains an interesting open problem to bridge the gap between the upper and lower bound.

For both our lower bounds we assume that the nodes have access to an infinite public string of random bits. In Section D.2, this is the string which the nodes use to collectively agree on some neural network network $f$ with respect to some distribution on such networks with at most $H$ ReLU units.

### D.1 Lower Bound: Communication Complexity

We consider a forest graph $G$ composed of pieces $G_1, G'_1, G_2, G'_2 \ldots, G_m, G'_m$, for $m = \Theta(\sqrt{n})$. Each piece $G_k$ consists of a "top" node $u_k$, which is only connected to a "middle" node $v_k$, which in turn is connected to $k$ "bottom" nodes $w_{k,1}, \ldots, w_{k,k}$, and $G'_k$ is simply a duplicate of $G_k$ (with vertices $u'_k, v'_k$, and $w'_{k,j}$ for $1 \leq j \leq k$). See Figure 3 for a depiction of $G$. We note that after two rounds, each $u_k$ (and $u'_k$) should know the respective value of $k$, because the local graph of depth 2 around $u_k$ is distinct for each $k \in [m]$.

**Theorem D.1.** *Suppose there exists a public random string $r$ that every node of $G$ has access to, and each node additionally has some independent private randomness. Suppose there is a communication protocol where by the end, with probability at least $3/4$, the following hold.*

- *For every $k \in [m]$, the top nodes $u_k$ and $u'_k$ output the same value.*

- *For every $k \neq \ell \in [m]$, the top nodes $u_k, u_\ell$ output distinct values.*

*Then, there must be some $k$ such that the edge $(u_k, v_k)$ or the edge $(u'_k, v'_k)$ has at least $\Omega(\log n)$ total bits of communication. Hence, if there are only $O(1)$ rounds of communication, one of those rounds must have sent $\Omega(\log n)$ bits of communication across the edge.*

**Remark D.2.** *In comparison to Definition 1.1, we note that our lower bound works for $p = \frac{1}{4m^2} = \Theta\left(\frac{1}{n}\right)$ in Definition 1.1. This is because we require all nodes $u_k, u_\ell$ to simultaneously have different outputs with probability at least $\frac{3}{4}$, which is implied by a union bound if every $u_k, u_\ell$ for $k \neq \ell$ have different outputs with probability at least $1 - \frac{1}{4m^2}$. In addition, we actually prove a stronger lower*

*bound against Definition 1.1, because our lower bound holds even if we allow each node to have its own independent private randomness, and only requires nodes with the same local neighborhood to output the same answer simultaneously with probability* $3/4$ *instead of probability* $1$.

*Proof.* First, we note that we may assume the communication is one-way from $v_k$ to $u_k$. This is because the node $v_k$ can simulate all communication from $u_k$, as $u_k$ has no information about neighbors apart from $v_k$. So, we just need to show the one-way communication complexity is $\Omega(\log n)$. Next, we will assume there is no public randomness - we will remove this assumption at the end. So, each $u_k$ (resp., $u'_k$) receives at most $b$ bits of information from $v_k$ (resp., $v'_k$). If $v_k$ sends a randomized message of length $b$ to $u_k$ and $u_k$ uses this message to produce some output $o_k$, with probability at least $\frac{3}{4}$ the outputs $o_k$ must all be distinct. In addition, for each $k$, the outputs of the duplicate copies of $u_k$ must be the same with probability at least $\frac{3}{4}$. Our goal is to show that $b = \Omega(\log n)$.

Let $f$ be a randomized function from $[m]$ to $\{0, 1\}^b$, representing the randomized message $v_k$ sends to $u_k$ assuming it has full knowledge of its number of neighbors. Let $g$ be a randomized function from $\{0, 1\}^b$ to some arbitrary output space $\mathcal{O}$, which represents the final output of $u_k$ after it has seen the message from $v_k$. Then, for all $k \neq \ell \in [m]$, $\mathbb{P}[g_1(f_1(k)) \neq g_2(f_2(\ell))] \geq \frac{3}{4}$, but for all $k \in [m]$, $\mathbb{P}[g_1(f_1(k)) = g_2(f_2(k))] \geq \frac{3}{4}$. Here, $g_1, g_2$ represent the function $g$ with different instantiations of the randomness, and $f_1, f_2$ represent the function $f$ with different instantiations of the randomness.

Fix some $k \in [m]$ and for each output $o \in \mathcal{O}$, let $p_k(o)$ represent the probability of outputting $g(f(k)) = o$. Note that $p_k(o)^2$ is the probability that $u_k$ outputs $o$ *and* $u'_k$ outputs $o$, so the probability that $u_k$ and $u'_k$ have the same output equals $\sum_{o \in \mathcal{O}} p_k(o)^2$, which we are assuming is at least $\frac{3}{4}$. This means that $\max_{o \in \mathcal{O}} p_k(o) = \sum_{o \in \mathcal{O}} p_k(o) \cdot \max_{o \in \mathcal{O}} p_k(o) \geq \sum_{o \in \mathcal{O}} p_k(o)^2 \geq \frac{3}{4}$. Therefore, for all $k \in [m]$, there exists an output $\tilde{o}_k$ such that $\mathbb{P}[g(f(k)) = \tilde{o}_k] \geq \frac{3}{4}$. Therefore, there must exist a value $s_k \in \{0, 1\}^b$ such that $\mathbb{P}[g(s_k) = \tilde{o}_k] \geq \frac{3}{4}$. Indeed, if not, then for any distribution over $s_k \in \{0, 1\}^b$, we have that $\mathbb{P}[g(s_k) = \tilde{o}_k] < \frac{3}{4}$, which means that $\mathbb{P}[g(f(k)) = \tilde{o}_k] < \frac{3}{4}$. In addition, $\tilde{o}_k$ is different across all $k$, as if $\tilde{o}_k = \tilde{o}_\ell$ for some $k \neq \ell$, then $\mathbb{P}[g(f(k)) = g(f(\ell)) = \tilde{o}_k] \geq \left(\frac{3}{4}\right)^2 \geq \frac{1}{2}$, which means that $\mathbb{P}[g(f(k)) \neq g(f(\ell))] \leq \frac{1}{2}$.

But now, note that the $s_k$'s must be distinct, because if $s_k = s_\ell$, then because $\mathbb{P}[g(s_k) = \tilde{o}_k] \geq \frac{3}{4}$ and $\mathbb{P}[g(s_\ell) = \tilde{o}_\ell] \geq \frac{3}{4}$, this means that $\tilde{o}_k = \tilde{o}_\ell$. Therefore, $s_1, \ldots, s_m$ are all distinct. But since each $s_i$ lies in $\{0, 1\}^b$, this means that $b = \Omega(\log m) = \Omega(\log n)$, as desired.

To finish, we revisit the fact that we assumed there was no public randomness. Let us reintroduce the random string $r$ that every node of $G$ is given. We assume that with probability at least $3/4$, the top nodes $u_k, u'_k$ have the same output for all $k \in [m]$ and that the nodes $u_1, \ldots, u_m$ output pairwise distinct values. But as this event happens with probability at least $3/4$ over a random string $r$, there must exist a choice of $r$ for which it happens with probability at least $3/4$ conditioned on $r$. But then we are back to the case where there is no public randomness, as desired. $\square$

### D.2 Lower Bound: ReLU Units

We next prove a lower bound on the number of ReLU units that we need to implement the WL test as a graph neural network. To do so, we will use that any neural network $\phi : \mathbb{R}^t \to \mathbb{R}^\ell$ which uses ReLU's as activation functions induces a partition of $\mathbb{R}^t$ into convex polytopes $R_1, \ldots, R_N$ such that $\phi$ restricted to each $R_i$ is just a linear function. Moreover, the number of such regions can be bounded using the following theorem.

**Theorem D.3** (Proposition 4 in [MPCB14]). *The number of linear regions of any ReLU network* $\phi : \mathbb{R}^t \to \mathbb{R}^d$ *with a total of $H$ ReLU units is at most $2^H$.*

Let $t, \ell, F \in \mathbb{N}$ be given and denote by $[F] = \{0, 1, \ldots, F - 1\}$. Let $\Phi_H$ denote the family of neural networks $\phi : \mathbb{R}^t \to \mathbb{R}^\ell$ with at most $H$ ReLU units mapping. We will consider the following model (see Figure 4): The algorithm designer picks a distribution $D$ over $\Phi_H$. Let $G$ be an arbitrary $n$-node graph and recall that $\mathcal{N}(i)$ denotes the neighborhood of node $i$ (including $i$ itself). For arbitrary inputs $x_1, \ldots, x_n \in [F]^t$ to the $n$ nodes (which we think of as the sums of the messages that the

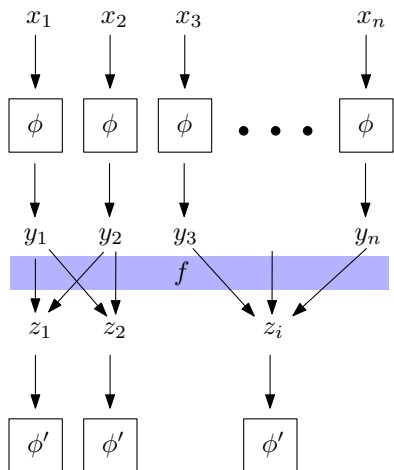

Figure 4: The structure of the GNN in our lower bound model. For every node $i$, node $i$ receives the message $x_i$ which is the sum of the labels of the neighbors of $i$. The algorithm then picks a random $\phi \in \Phi_H$ with respect to $D$ and for each $i = 1, \ldots, n$, calculates $y_i = \phi(x_i)$ which is the new label of node $i$. Next, each node $i$, sends $y_i$ to each of its neighbors and every node $i$ then calculates $z_i = f(\{y_j \mid j \in \mathcal{N}(i)\}) = \sum_{j \in \mathcal{N}(i)} y_j$. The $z_i$'s then form the node inputs at the next iteration to a new randomly chosen $\phi' \in \Phi_H$.

nodes receive), the GNN operates as follows: Using the publicly available random string, the nodes collectively pick a random neural network $\phi \in \Phi_H$ with respect to $D$. For $i = 1, \ldots, n$, node $i$ then calculates $y_i = \phi(x_i)$ (which we think of as the new label of node $i$). Then each node $i$ sends $y_i$ to each node in its neighborhood $\mathcal{N}(i)$ and calculates $z_i = \sum_{j \in \mathcal{N}(i)} y_j$. We remark that this notation is different from what we introduced in Section 1.1 and 1.2. Referring back to those sections, $i$ corresponds to node $u$, $x_i$ corresponds to $\mathcal{S}_u^{(k-1)}$, i.e., the sum of the received messaged of node $u$ in iteration $k-1$, $\phi$ corresponds to $\phi^{(k)}$, $y_i$ corresponds to $h_u^{(k)}$, i.e., the new label of node $u$, and $z_i$ corresponds to $\mathcal{S}_u^{(k)}$, i.e., the sum of the received messaged of node $u$ in iteration $k$. We have made this switch in notation to make the argument that follows less unwieldy.

We would like our GNN to satisfy that for any $n$-node graph $G$, and arbitrary inputs $x_1, \ldots, x_n$ to the nodes, it holds with probability at least $9/10$ over the randomness of $D$ that $z_i \neq z_j$ for all $i, j$ such that the multisets $\{x_k \mid k \in \mathcal{N}(i)\}$ and $\{x_k \mid k \in \mathcal{N}(j)\}$ are different. The following theorem provides a lower bound on the number of ReLU units $H$ needed for this property to hold.

**Theorem D.4.** *Suppose that the neural networks in $\Phi_H$ have at most $H \leq \lg F - 4$ ReLU units. Then there exists a graph $G$ on $n$ nodes and inputs $x_1, \ldots, x_n \in [F]^t$ such that if $\mathcal{N}(1), \ldots, \mathcal{N}(n)$ are the neighborhoods of the nodes of $G$, then with probability at least $1 - \left(\frac{3}{4}\right)^{n/6}$, there exists $i, j \in [n]$ such that $z_i = z_j$ even though the multisets $\{x_k \mid k \in \mathcal{N}(i)\}$ and $\{x_k \mid k \in \mathcal{N}(j)\}$ are different. Thus, to simulate the WL test with neural networks from $\Phi_H$, we need $H > \log F - 3$ ReLU units.*

Before proving the theorem, we first explain how to interpret it as a lower bound for the computational complexity of implementing a WL iteration as in Definition 1.1 as a neural network. As an initial observation, note that in any iteration $k$, if for two nodes $u$ and $v$, the sums $S_u^{(k-1)}$ and $S_v^{(k-1)}$ are distinct (recall that $\mathcal{S}_u^{(k-1)} = \sum_{u' \in \mathcal{N}(u)} h_u^{(k-1)}$), then for the WL iteration to be successful, we must also have that $h_u^{(k)} \neq h_v^{(k)}$. This is because, $\mathcal{S}_u^{(k-1)} \neq \mathcal{S}_v^{(k-1)}$ implies that the multisets $\{h_{u'}^{(k-1)} \mid u' \in \mathcal{N}(u)\}$ and $\{h_{v'}^{(k-1)} \mid v' \in \mathcal{N}(v)\}$ are also distinct. But then Definition 1.1 yields that we need $h_u^{(k)} \neq h_v^{(k)}$ (at least with some probability $1 - p$). Now, consider two nodes $i = u$ and $j = v$ such that in some iteration of the WL test, the multisets of sums $\{\mathcal{S}_{u'}^{(k-1)} \mid u' \in \mathcal{N}(u)\}$ and $\{\mathcal{S}_{v'}^{(k-1)} \mid v' \in \mathcal{N}(v)\}$ are different. This corresponds to the multisets $\{x_k \mid k \in \mathcal{N}(i)\}$ and $\{x_k \mid k \in \mathcal{N}(j)\}$ being different. We would like to argue that for the WL iteration to be successful

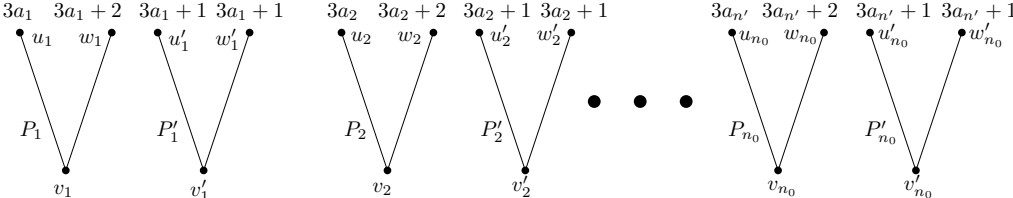

Figure 5: The graph of our lower bound construction of Theorem D.4. It consists of $n/3$ copies of a path of length 3 grouped into pairs of two. For each such pair of paths $(P, P')$, the two end nodes of $P$ are assigned (random) inputs (which we think of as the sum of their received messages) that are guaranteed to have the same sum as the sum of the inputs to the end nodes of $P'$. The idea of the proof is that if the neural network $\phi$ has too few ReLU unit, then this linear dependence is preserved with probability $\Omega(1)$ even after applying $\phi$.

according to Definition 1.1, for each such pair of nodes $u, v$, we must have that also $\mathcal{S}_u^{(k)} \neq \mathcal{S}_v^{(k)}$ with some good probability (the sums of labels in the next iteration differ). Theorem D.4 tells us that the probability of this happening is very low if we use too few ReLU units. Now why do we require that $\mathcal{S}_u^{(k)} \neq \mathcal{S}_v^{(k)}$ for such a pair of nodes $u, v$?

Since the multisets $\{\mathcal{S}_{u'}^{(k-1)} \mid u' \in \mathcal{N}(u)\}$ and $\{\mathcal{S}_{v'}^{(k-1)} \mid v' \in \mathcal{N}(v)\}$ are different, by the initial observation, the multisets $\{h_{u'}^{(k)} \mid u' \in \mathcal{N}(u)\}$ and $\{h_{v'}^{(k)} \mid v' \in \mathcal{N}(v)\}$ must also be distinct for the WL test to be successful. But since the multisets of labels $\{h_{u'}^{(k)} \mid u' \in \mathcal{N}(u)\}$ and $\{h_{u'}^{(k)} \mid v' \in \mathcal{N}(v)\}$ are distinct it follows by another application of Definition 1.1, that we must also have that $h_u^{(k+1)} \neq h_v^{(k+1)}$. However, the only way this can happen is if $\mathcal{S}_u^{(k)} \neq \mathcal{S}_v^{(k)}$ as otherwise these two sums will be mapped to the same label by $\phi^{(k+1)}$.

We remark that this lower bound applies to an isolated WL iteration rather than a full sequence of iterations. In particular, the inputs $x_1, \ldots, x_n \in [F]^t$ (corresponding to the sums $\{\mathcal{S}_u^{(k-1)} \mid u \in G\}$) are adversarially chosen while in reality these inputs are not arbitrary but are the result of a prior WL iteration. Our construction in Section 3 indeed works against such adversarially chosen sums in the sense that different multisets of sums are mapped (via applying $\phi^{(k)}$ and summing the outputs for each multisets) to different sums with high probability, and as such our lower bound is exactly a lower bound for this harder problem. However, in general the sums $S_v^{(k-1)}$ are not adversarially chosen, and it would very be interesting to find a lower bound that does not require this assumption but works all the way from a graph and its initial labels.

*Proof of Theorem D.4.* We exhibit a graph $G$ and a distribution $D_0$ over the possible inputs, such that for any neural network $\phi : [F]^t \to \mathbb{R}^\ell$ with at most $H$ ReLU units, if $(x_1, \ldots, x_n) \in ([F]^t)^n$ is chosen with respect to $D_0$, then with probability at least $1 - \left(\frac{3}{4}\right)^{n/6}$, there exists $i, j$ such that $z_i = z_j$ even though the multisets $\{x_k \mid k \in \mathcal{N}(i)\}$ and $\{x_k \mid k \in \mathcal{N}(j)\}$ are different. It then follows from Yao's minimax principle that for any distribution $D$ over $\Phi_H$, there exists an input $x = (x_1, \ldots, x_n)$ such that the same bad event occurs with the same high probability. As desired.

Let us start out by describing the graph $G$. Assume with no loss of generality that 6 divides $n$, i.e. $n = 6n_0$ for some natural number $n_0$. The graph $G$ simply consists of $2n_0$ copies of a path of length 2 grouped into pairs $(P_1, P_1'), \ldots, (P_{n_0}, P_{n_0}')$ (see Figure 5). Denote the vertices of path $P_i$ by $(u_i, v_i, w_i)$ and similarly the vertices of path $P_i'$ by $(u_i', v_i', w_i')$. We next proceed to describe the distribution $D_0$ over inputs $(x_1, \ldots, x_n) \in ([F]^t)^n$. Here, for each $i < n_0$, we let $x_{3i}, x_{3i+1}, x_{3i+2}$ be the inputs to nodes $u_i$, $v_i$ and $w_i$ respectively and $x_{3i+3}, x_{3i+4}, x_{3i+5}$ be the inputs to nodes $u_i'$, $v_i'$ and $w_i'$. To do so, let $\ell = \lfloor F/3 \rfloor$ and define for each $0 \leq a < \ell$ and $b \in [F]^{t-1}$ the set $F_{a,b} = \{(3a, b), (3a+1, b), (3a+2, b)\}$. Thus, $F_{a,b} \subseteq [F]^t$ consists of the three vectors which have their first coordinate equal to respectively $3a, 3a+1$ and $3a+2$, and $b$ as their last $t-1$ coordinates. For each $i < n_0$, we independently pick uniformly random $0 < a \leq \ell$ and $b \in [F]^{t-1}$. We then define $x_{6j} = (3a, b)$, $x_{6j+2} = (3a+2, b)$, and $x_{6j+3} = x_{6j+5} = (3a+1, b)$ (see Figure 5). We further define $x_{6j+1} = x_{6j+4} = 0$ (the exact value is unimportant, as long as they are equal). Note that $x_{6j} + x_{6j+1} + x_{6j+2} = x_{6j+3} + x_{6j+4} + x_{6j+5}$. In particular, if the random $\phi \in \mathcal{F}$ was a linear

map, we would have that

$$z_{6j+1} = \phi(x_{6j}) + \phi(x_{6j+1}) + \phi(x_{6j+2}) = \phi(x_{6j+3}) + \phi(x_{6j+4}) + \phi(x_{6j+5}) = z_{6j+4}, \quad \text{(D.1)}$$

in spite of the multisets, $\{x_{6j}, x_{6j+1}, x_{6j+2}\}$ and $\{x_{6j+3}, x_{6j+4}, x_{6j+5}\}$ being different. Now, $\phi$ is a neural network, so this identity does not need to hold. The idea is however, that if $\phi$ has only few ReLU units, then we obtain a good upper bound on the number of *linear regions* by Theorem D.3 and since $a$ and $b$ are random, the set $F_{a,b}$ is likely to be fully contained in one of these regions. And since $\phi$ restricted to this region in linear, (D.1) holds in this case.

To formalize this, assume that $\phi : [F]^t \to \mathbb{R}^\ell$ is a neural network in $\Phi_H$ with at most $H$ ReLU units. Using Theorem D.3, it follows that $[0, F-1]^t$ can then be partitioned into at most $2^H$ convex regions $R_1, \ldots, R_N$, such that for each region $R_i$, it holds that $\phi$ restricted to $R_i$ is simply a linear function. For $0 \le a < \ell$ and $b \in [F]^{t-1}$, we let $L_{a,b}$ be the straight line segment connecting the two points $(3a, b)$ and $(3a+2, b)$. By convexity, for a fixed $R_i$ and a fixed $b \in [F]^{t-1}$, at most 2 of the line segments in $\{L_{a,b} \mid 0 \le a < \ell\}$ can intersect the boundary of $R_i$. Since there are $F^{t-1}$ distinct choices for $b$ and $N \le 2^H$ choices for $i$, this implies that the number of line segments $L_{a,b}$ that can cross *any* of the boundaries of the convex regions is at most $2^{H+1}F^{t-1}$. Each of the remaining segments $L_{a,b}$ must be fully contained in some region $R_i$. There are $\lfloor F/3 \rfloor F^{t-1}$ choices of $a$ and $b$ in total, and thus, at least $\lfloor F/3 \rfloor F^{t-1} - 2^{H+1}F^{t-1}$ of the segments $L_{a,b}$ are fully contained in one of the regions $R_i$, which means that $\phi$ restricted to $L_{a,b}$ acts as a linear map. For such a segment $L_{a,b}$, we get by linearity that

$$\phi(3a, b) + \phi(3a+2, b) = 2\phi(3a+1, b).$$

In other words, if for a fixed $i \le n_0$, $(a, b)$ is chosen such that $L_{a,b}$ is fully contained in one of the convex regions, then (D.1) is satisfied. It follows that for any given $i < n_0$,

$$\Pr[z_{6i+1} = z_{6i+4}] \ge 1 - \frac{2^{H+1}F^{t-1}}{\lfloor F/3 \rfloor F^{t-1}} \ge 1 - \frac{12 \cdot 2^H}{F}. \quad \text{(D.2)}$$

Since the event $(z_{6i+1} = z_{6i+4})_{i \le n_0}$ are independent (as we choose independent $(a, b)$ for each pair of paths $(P_i, P_i')$), it follows that

$$\Pr[\exists 0 \le i \le n_0 : z_{6i+1} = z_{6i+4}] \ge 1 - \left(\frac{12 \cdot 2^H}{F}\right)^{n_0}.$$

If in particular, $H \le \lg F - 4$, we obtain that

$$\Pr[\exists 0 \le i \le n_0 : z_{6i+1} = z_{6i+4}] \ge 1 - \left(\frac{3}{4}\right)^{n/6},$$

As the multisets $\{x_k \mid k \in \mathcal{N}(6i+1)\}$ and $\{x_k \mid k \in \mathcal{N}(6i+4)\}$ are different, this completes the proof. $\square$

**Remark D.5.** *In analogue with our construction in Section 2 and Section 3, we assumed in the above proof that the aggregate function $f$ is the summation function. As such, $f$ is just another neural network but without a single ReLU unit. We can therefore think of the combined computation performed by $f$ and $\phi$ (illustrated in Figure 5) as the result of applying a single neural network. It follows from this observation (and the proof of Theorem D.4) that in the more general setting where $f$ is a function in $\Phi_{H_1}$ and where the neural network $\phi \in \Phi_{H_2}$, then we must have that $H_1 + H_2 > \log F - 4$ in order to successfully simulate the WL test.*

### D.2.1 Better bounds on the number of linear regions.

The proof of Theorem D.4 used that the number of convex linear regions of any neural network with at most $H$ ReLU's is at most $2^H$. However, in many cases one can obtain better upper bounds on the number of such regions, and this directly translates to a better lower bound than the one given in Theorem D.4. Indeed, if the family of neural networks $\Phi$ satisfies that the domain of any $\phi \in \Phi$ can be partitioned into at most $K$ convex regions such that $\phi$ restricted to each of these regions is linear, then the lower bound in (D.2) instead becomes

$$\Pr[z_{6i+1} = z_{6i+4}] \ge 1 - \frac{12 \cdot K}{F}.$$

In particular, when using independence of the events $(z_{6i+1} = z_{6i+4})_{i<n_0}$, we just need $K \leq \frac{F}{24}$, say, to get that an error occur with probability at least $1 - 2^{-\Omega(n)}$. Plugging in the bound $K \leq 2^H$ of Theorem D.3 gave the desired bound of Theorem D.4 which led to the $\Omega(\log F)$ lower bound. If we instead use the more fine grained theorem below, we obtain better bounds for shallow neural networks with low input dimension as stated in Corollary D.7.

**Theorem D.6** (Theorem 1 in [RPK$^+$17])**.** *Any ReLU neural network with input dimension $t$, width $w$, and depth $d$ has at most $O(w^{d \cdot t})$ linear regions.*

**Corollary D.7.** *Let $\Phi_{t,d,w}$ consist of all ReLU neural networks with input dimension $t$, depth $d$, and width $w$. Suppose that we are in the setting of Theorem D.4, except that the neural network is picked from $\Phi_{t,d,w}$. Then the conclusion of the theorem holds as long as $w^{d \cdot t} \leq cF$ for a small enough constant $c$. In particular, to simulate the WL test with neural networks from $\Phi_{t,d,w}$, we need $dw = \Omega(dF^{\frac{1}{dt}})$ ReLU units.*

As an example, for shallow neural networks with low input dimension, say with $d, t = O(1)$, this lower bound becomes $dw = F^{\Omega(1)}$, i.e. polynomial rather than logarithmic in the size of the underlying field.

For certain architectures of the neural networks one can obtain even stronger bounds on the number of linear regions (see e.g., Proposition 3 in [Mon17] and Theorem 1 in [STR18]). These bounds are parametrized in the number of ReLU units in each of the $d$ layers of the neural networks. One can therefore obtain even more fine grained lower bounds on the number of ReLU units if one makes more assumptions on the family of neural networks but the bounds are more opaque and we refrain from stating them here.

### D.2.2 Description complexity

In this subsection, we consider more general function classes $\Phi$ that do not necessarily have to consist of neural networks. We prove that for any aggregation function $f$, if for any two distinct multisets of labels each of size at most $n$, there exists a function $\phi \in \Phi$ such that the multisets are mapped to different labels by $\phi \circ f$, then $|\Phi| = \Omega\left(\frac{n}{\log n}\right)$. It follows that the description complexity of $\Phi$ must be $\Omega(\log n)$. Our bound is combinatorial, and does not employ the linear structure of $F^t$. Hence we may just put $t = 1$ and think of $F$ as a set rather than a vector space.

**Theorem D.8.** *Let $F$ and $n$ be natural numbers and let $N = \sum_{i=0}^{n} \binom{i+F-1}{i}$ be the number of multisets of $[F]$ of size at most $n$. Suppose that $f$ is any aggregation function mapping multisets of $[F]$ to some range $\mathcal{R}$. Let $\Phi$ be any set of functions from $\mathcal{R}$ to $F$ and assume that $|\Phi| < \frac{\log N}{\log F}$. Then there exists distinct multisets $A$ and $B$ each with at most $n$ elements from $[F]$ such that $\phi(f(A)) = \phi(f(B))$ for all $\phi \in \Phi$.*

*Proof.* Note that each function $\phi \in \Phi$ induces a partition $\mathcal{P}_\phi$ on the set of these multisets induced by the equivalence relation defined by $X \sim_\phi Y \iff \phi(f(X)) = \phi(f(Y))$. By repeated application of the pidgeonhole principle, there must exists a collection $\mathcal{C}$ of multisets each containing at most $n$ elements from $[F]$ such that (1) for all $\phi \in \Phi$ and all $X, Y \in \mathcal{C}$, $X \sim_\phi Y$ and (2) $|\mathcal{C}| \geq N/F^{|\Phi|}$. If in particular $|\Phi| < \frac{\log N}{\log F}$, we must have that $|\mathcal{C}| \geq 2$. Letting $A$ and $B$ be distinct elements of $\mathcal{C}$, we have that $\phi(f(A)) = \phi(f(B))$ for all $\phi \in \Phi$. $\qed$

Note that number of distinct degrees of the vertices of a simple $n$-node graph could be as large as $\Omega(n)$, so it is a natural assumption that also $F = \Omega(n)$. Indeed, if $F$ is smaller, then for such a graph, the simulation of the WL test will fail with probability 1 since it must inevitably assign two nodes of distinct degrees the same label in $F$. With this assumption, it follows that

$$\log N = \log \sum_{i=0}^{n} \binom{i+F-1}{i} \geq \log \binom{n+F-1}{n} \geq \log \left(1 + \frac{F-1}{n}\right)^n$$
$$= \Omega\left(n \cdot \max\left\{1, \log \frac{F}{n}\right\}\right),$$

using the inequality $\binom{n}{k} \geq \left(\frac{n}{k}\right)^k$. Thus,

$$\frac{\log N}{\log F} = \Omega\left(n \cdot \frac{\max\left\{1, \log \frac{F}{n}\right\}}{\log F}\right) = \Omega\left(\frac{n}{\log n}\right).$$

In particular, the description complexity of $\Phi$ has to be $\Omega(\log n)$ in order to separate any two distinct multisets $X, Y \subset F$ each consisting of at most $n$ elements. We note that the description complexity of the construction in Section 3 is poly $\log n$.