# OpenReview forum: "Exponentially Improving the Complexity of Simulating the Weisfeiler-Lehman Test with Graph Neural Networks"
_NeurIPS.cc/2022/Conference — NeurIPS 2022 Accept_

### Official Review · Reviewer_Kh1P · 2022-07-08

**Rating:** 4
**Confidence:** 4
**Soundness:** 4 excellent
**Presentation:** 3 good
**Contribution:** 1 poor

**Summary:**

This paper proposes a collection of GNN constructions that can simulate the Weisfeler-Leman algorithm with high probability. Prior works showed the equivalence between the WL algorithm and GNNs using mainly the universal approximation theorem for MLPs, in order to prove that MLPs can be used to approximate injective message and update functions, so as to simulate the hash functions used in the WL algorithm. However, as the authors of the present manuscript note, this construction might in the worst case necessitate an exponential number of parameters for the MLP. To this end, the authors devise a special GNN construction, that can simulate the hash functions using $O(poly(logn))$ random parameters (as a warm-up they show a simple construction that requires a $O(poly(n))$ number of parameters) and $O(logn)$ bits of precision, and will succeed with probability at least $1 - 1/O(poly(n))$. Finally, the above results are complemented with lower bounds: the authors construct a graph, for which any communication protocol requires at least $\Omega(logn)$ bits of precision to simulate WL, while a $\Omega(logn)$ lower bound is also given for the number of ReLU units.


**Questions:**

- What does the lower bound on the number of the relu units imply about the overall number of parameters?
- Perhaps a relevant work (involving communication capacity) that the authors might want to discuss is the following:
Loukas, "How hard is to distinguish graphs with graph neural networks?", NeurIPS’20
- One suggestion to better motivate the paper, would be to include a small experimental section. This may include any potential shortcomings of GNNs to simulate WL, dependence on the number of parameters, scaling with the number of vertices, contrasting between random and learned GNNs etc. Also, here it might be interesting to observe if GNNs are actually simulating WL, or if they simply manage to distinguish the same non-isomorphic pairs.
- Is it possible that the ability to simulate WL correctly can be a useful inductive bias  (e.g., in order to counteract the oversquasing phenomenon) and lead to better generalising GNNs (this might be also an interesting motivation)?

- Minor: L41: MFK21 → MRF+19


**Limitations:**

- As I mentioned above, a discussion about the broader scope, including the limitations (especially about how this theoretical construction can be useful in practice) is missing.
- This is a purely theoretical paper - no potential negative societal impact can be foreseen.

**Strengths And Weaknesses:**

# Strengths

- This paper devises a intricate construction of a GNN that can simulate the WL algorithm with an exponentially smaller size than previous constructions.  From a theoretical standpoint this result is interesting since it shows that GNNs can be easily used as a fast heuristic for graph isomorphism (replacing WL), without the need for large parameter sizes in order to obtain strong guarantees of validity.
- Compared to previous works that simply used the universal approximation theorem to prove that GNNs can have injective message and update functions, this paper uses a special construction to approximate hash functions, in order to reduce the required number of parameters, which might be of independent theoretical interest. Of independent interest might also be the constructions for the lower bounds, which provide further insights.
- Although the paper is mostly directed to an audience versed in complexity theory and involves complicated concepts that the ML audience will be less familiar with, the authors have put a good effort to present their results in a clear and concise way, so that the interested reader can understand the high-level message.

# Weaknessess

- Most of my important concerns regarding this paper have to do with its relevance and practical significance. Although, I do believe that the constructions are theoretically interesting per se, it is unclear to me how this paper might benefit the GNN community and what insights we can gain from it. My impression is that the presentation of the theoretical results is kind of “dry” and the broader scope of the paper is unclear. In my opinion, a discussion is very much missing from the paper in order to better place it in the GNN context and explain its benefits to the community. I have several questions w.r.t. the above:
- The premise of the paper has to do with exactly simulating the WL algorithm. Could the authors explain why this is important per se? My intuition is that what we are usually mostly interested in is being able to distinguish the same non-isomoprhic graphs as the WL test, rather than simulating its exact iterations. Is it possible that we can obtain better bounds in this scenario? Are there cases where we will fail to distinguish a pair of graphs unless exactly simulating the WL algorithm?
- The paper presents worst case results. However, it has been widely observed in the GNN community that even random GNNs of small size (definitely not exponential) can be used in order to distinguish non-isomorphic, WL distinguishable, pairs of graphs. This naturally leads to the following question: in which circumstances are the results of this paper relevant? Is it the case that for real-world graphs (or for known graph distributions, e.g., Erdos-Renyi) we can obtain even better bounds? I believe that such a question would have been much more influential across the community.
- If I am not mistaken, throughout the paper, the authors use integer representations. Is it possible that the results can be improved with continuous representations (after all this is what we use in practical NNs)?
- Finally, it is my impression that this construction needs to be enforced in a NN and cannot be learned, which kind of defeats the purpose of having a NN in the first place. This is because, either the functions to be implemented are hard to learn with SGD (e.g., the mod function) or are not even differentiable (e.g., the indexing in step 5 L270).

---

> ### Author Response · Authors · 2022-08-02
> **Response to Reviewer Kh1P**
>
> Thank you very much for your valuable feedback. Below are our responses to your questions/concerns.
>
>
> **Importance of exactly simulating the steps of the WL-algorithm:** For distinguishing non-isomorphic graphs, it is a priori not important to simulate the individual steps of the WL-test. Indeed, it would be very exciting to see efficient improvements over the WL-test which can distinguish larger classes of non-isomorphic graphs. This is however a longstanding open question and beyond the scope of this paper.
>
> We highlight that within the context of GNNs, it makes sense to exactly simulate the WL-test since it follows from the results of Xu et al. and Morris et al. that this task characterizes the power of GNNs. In other words, while there might exist efficient algorithms for distinguishing more general classes of graphs than the WL-test (which would indeed be very exciting), we can not expect this from GNNs. Our results instead show that even very simple GNNs can have the power of the WL-test.
>
>
> **Better results for random/real-world graphs:** We wholeheartedly agree with the reviewer that GNNs of small size are powerful; indeed, that is the central message of our paper! We show that extremely efficient GNNs (with message size **logarithmic** and number of hidden units poly **logarithmic** in the size of the graph) are expressible enough to distinguish isomorphic / non-isomorphic graphs as well as the WL algorithm, an algorithm that characterizes the power of GNNs, as per Xu et al and Morris et al. The fact that we can get worst-case bounds for such small GNNs implies that these same bounds hold for *any* distribution of interest or *any* real-world graphs.
>
> Studying the behavior of the WL algorithm for special graph distributions is an interesting future direction. However, we remark that since our construction size is logarithmic in the message size and poly logarithmic in the number of hidden units, any specialized result would only improve over our bounds by a poly logarithmic factor, whereas our bound improves over the prior state of the art in Xu et al. and Morris et al. by exponential and polynomial factors respectively while having the benefit of being oblivious (meaning we use the *same* network construction for all graphs). Thus our result can be seen as filling in an important gap between the theoretical understanding of the expressibility capabilities of GNNs and the practical observation that small GNNs are sufficiently powerful for distinguishing non-isomorphic, WL distinguishable, pairs of graphs.
>
>
> **Integer representations:** Since our upper bounds are already very strong (O(log n) bits and dimensions),  there is not much room for further improvement. However, it is an interesting question if one could go below log n dimensions, while keeping the neural networks small.
>
>
> **Learning:** Our focus is on representation capabilities of GNNs as opposed to their learnability. We believe this is of broad interest to the ML community as such results have had significant impacts on machine learning, both in theory and practice. Please see our response titled “Meaning of our result to ML researchers/learnability” to Reviewer j9Sq for further discussion. However, we note that our constructions can be implemented using standard ReLU neural networks, which are differentiable. In particular, the indexing subroutine can be implemented as in Corollary 3.3; it gives a construction of a ReLu network which accepts an index and outputs the appropriate coordinate of an eps-biased random vector. Please note that the modulo and indexing functions are not built into the architecture, and therefore their (lack of) differentiability does not affect the learning process. Instead, the architecture consists only of standard (and differentiable) linear layers and ReLU units, and if the network was to be trained, then gradients would only need to flow through those standard layers. The modulo and indexing functions only arise as hindsight interpretations to certain parameter settings of those standard layers.
>
>
> **Lower bound implications on number of parameters:** Of course, the number of parameters depends not just on the number of ReLU units, but also the width/depth tradeoff of the network (with the number of parameters in each layer corresponding to the product of the number of units in the two adjacent layers) as given in section D.2.1. Naively, the number of parameters is at least the number of ReLU units. To give some concrete examples (assuming for simplicity constant input dimension), for a shallow neural network with depth d = O(1), the width of each layer must be w=Omega(n), so the total number of parameters will be O(d n^2) = O(n^2). For a deeper neural network with  d = O(log(n)), the width must be constant w = Omega(1), and the total number of parameters is O(log(n)).

---

> > ### Author Response · Authors · 2022-08-02
> > **Response to Reviewer Kh1P (ctd.)**
> >
> > **Loukas NeurIPS’20:** Thanks for the reference! From our understanding, the notion of isomorphism which the GNNs have to compute in this paper is significantly stronger than that studied in Xu et al and Morris et al. In Loukas, the graph neural network has to compute the “isomorphism class” of its input graph which is a unique symbol that distinguishes it from all non-isomorphic graphs. The lower bounds in this paper then ultimately comes from the combinatorial count of all possible non-isomorphic graphs/trees on n nodes. Conversely, Xu et al. and Morris et al. study isomorphism testing between a *pair* of graphs with the guarantee that the GNN outputs are the same if WL can distinguish the two graphs and the outputs are different otherwise. Our results with inverse polynomial failure probability are also in this setting. However, because we model the problem in terms of successful WL iterations, we should be able to extend them to the setting of Loukas! By instead using inverse exponential failure probability, we can union bound over all pairs of non-isomorphic (as distinguished by WL) graphs. This would blow up the message size and number of parameters to polynomial rather than polylogarithmic, but this corresponds with Loukas’ lower bounds which imply that if we want to distinguish all possible graphs (rather than a pair or small set of graphs), the messages must use Omega(n) bits (assuming the GNN is run for n iterations). We will add a citation to this paper along with this discussion.
> >
> >
> > **Experimental section:** Thank you for the suggestion. We are designing an experiment to substantiate the empirical advantage of our theoretical analysis (see also response to Reviewer epxq) which will be added to the revised version of our paper.
> >
> >
> > **Inductive bias:** This is an exciting direction to explore. Indeed, it is possible that incentivizing GNNs to learn to simulate WL can lead to better utilization of their expressive power and improve performance, though it is not immediately clear how to incentivize this and induce the desired inductive bias. We leave this as an important direction for future work.

---

> ### Author Response · Authors · 2022-08-07
> **Follow up to Reviewer Kh1P**
>
> Dear Reviewer Kh1P,
>
> Did we address all your concerns satisfactorily? If your concerns have not been resolved, could you please let us know which concerns were not sufficiently addressed so that we have a chance to respond?
>
> Many thanks,
> The authors

---

### Official Review · Reviewer_epxq · 2022-07-09

**Rating:** 6
**Confidence:** 4
**Soundness:** 3 good
**Presentation:** 3 good
**Contribution:** 2 fair

**Summary:**

The authors provide a new theoretical GNN construction that is able to simulate a 1-WL test with fewer parameters and smaller messages, by borrowing some ideas from randomized algorithms.

**Questions:**

What issues do you perceive implementing this algorithm in practice? Or any learnability issues?

Could you provide a theoretical estimate (a plot) of when  the poor scaling of traditional GNNs (GIN) would start to matter (how many nodes should there be in a graph)  if we assume use of real-valued embedding vectors (and are generous w.r.t. what a GNN/NN can learn)? Essentially, could you estimate the constants hidden in the O notation for message size.

Of course what would really convince me is a practical implementation of the proposed construction and a synthetic experiment, which shows a better scaling of the proposed approach (at least with fixed weights but ideally learned ones) compared to a traditional GNN woth learned weights.

**Limitations:**

I do not perceive any negative societal impact and any limitations besides the drawbacks meantioned above.

**Strengths And Weaknesses:**

I really liked, that the authors decided to look at GNNs like a true algorithm as one does in theoretical computer science. I also liked the introduction of randomization and use of ideas from randomized algorithms. I think this can be a fruitful avenue for design of more efficient neural network architectures in general.
The writing was excellent and theoretical analysis is sound. I also appreciate the inclusion of the lower bounds.

However, the man problem of the paper is that it is very far removed from any practical considerations. Authors do not implement their GNN or do any experiments.
While their approach results in smaller message size (which is one of the main benefits of their approach) it's unclear to me how much it would matter in practice.
GNNs use real-valued embedding vectors, that even with modest vector lengths already provide us very large effective bit count. It's possible that we do not run into any message size problems for any real-world-sized graph (graphs used in graph classification are rarely larger than a few thousand nodes).

---

> ### Author Response · Authors · 2022-08-02
> **Response to Reviewer epxq**
>
> Thank you very much for your valuable feedback. Below are our responses to your questions/concerns.
>
>
> **Practical considerations regarding message size:** First, we emphasize that our construction (as well as lower bounds) address not only message size but also the number of parameters/units of the neural networks. In terms of message size, our improvement is not just in the number of bits in the message, but the dimension of the message vectors (output vectors of the neural network). In both Xu et al. and Morris et al, the message vectors are one-hot vectors of dimension n. On the other hand, our construction uses vectors of dimension O(log n). The fact that our message vectors are binary is an added bonus. So, even if we consider the case where each coordinate is a 64-bit float rather than a single bit, our construction constitutes a major improvement over prior work by showing that the connection between GNNs and the WL test remains even if the output dimension of the neural networks is on the order of log(n) as opposed to n. This has practical applications as GNNs in the wild often have relatively modest output dimensions even if each coordinate contains 64 bits.
>
>
> **Scaling comparison:** To be concrete, the dimension of the vectors in our construction is C log_{3/2} n for some choice of C which determines the failure probability. To be conservative, let’s say C = 4 to have failure probability at any given iteration for any pair of nodes is 1/n^4 . On the other hand, the dimension of the vectors in the prior constructions is exactly n. So, for instance, if n=1000, our construction requires message dimension d=68 while prior constructions require d=1000. Of course, this only improves as n grows as our construction is exponentially smaller.
>
>
> **Implications for real-world-sized graphs:** The message of our paper is meant to be conceptual rather than practical. In particular, we believe that gauging the true trade-off between the number of parameters and the model's expressibility is an important question and a worthwhile goal, even if it turns out that the previously known trade-offs, despite being loose and suboptimal, are sufficient for explaining currently used graph sizes of up to thousands of nodes. The demand for deep learning architectures (and GNNs in particular) that scale to ever larger datasets poses a major challenge and focus of research, and there are certainly larger graphs to be considered (for example, the SNAP dataset of large networks [https://snap.stanford.edu/data/index.html] contains many graphs with millions of nodes).
>
>
> **Implementing the construction:** We appreciate the suggestion to experimentally validate our theoretical results, and establish that our construction indeed leads to better expressivity than previous ones. This would shed more light on how to interpret our results. We are designing the experiment and will add it to the final version of our paper.

---

> > ### Comment · Reviewer_epxq · 2022-08-06
> > **Response to Authors**
> >
> > Thank you for your answers. I appreciate your efforts. Indeed some experiments would in my opinion make this paper 'whole'. It's a bit harder since we can't see them yet. But I'm cautiously optimistic as the rest of the paper is nice and have increased my score accordingly.
> >
> > As Reviewer Kh1P has highlighted, positioning the paper more carefully in context of the current GNN literature and distilling the message/contribution for the more practical folks would also help to improve the final verion.

---

### Official Review · Reviewer_j9Sq · 2022-07-10

**Rating:** 7
**Confidence:** 3
**Soundness:** 4 excellent
**Presentation:** 4 excellent
**Contribution:** 3 good

**Summary:**

The paper studies the complexity of simulating the WL test with message-passing neural networks. The fact that this can be done is well known since 2019, but an efficient computation was lacking. The main result of this paper is a construction of a neural network implementing the local update function in polylog(n) parameters using feature vectors of size lo O(log n) bits. This is significantly better than previous constructions but comes at a cost of a certain probability of failure since the construction relies on random weight initialization. In addition to this main result, the authors also propose three other results: (1) A simpler and less efficient construction (as the first step toward efficient architecture); and (2) two logarithmic lower bounds on the feature length and the size of the network.


**Questions:**



Relation to ML - the way this paper is currently written emphasizes the construction and its complexity. Since NeurIPS is conference on machine learning and other close fields, I feel it would be great to add a discussion on the meaning of this result to ML people: can we leverage this result in order to modify existing MPNNs? To propose new ones? How hard is it to learn weights leading to this construction (which is heavily based on discrete quantities)? How is the randomness related to the learning process? (to be fair, most papers that belong to this type of paper ignore my last questions)

– what can we say about deterministic construction? Are the lower any lower bounds?
Limitations

**Limitations:**

Properly addressed

**Strengths And Weaknesses:**

– The paper is very well written, especially for a theoretical paper. I enjoyed reading it.
– Interesting problem and results - the ability of MPNNs to simulate the WL test is a significant result in the GNN community. Nevertheless, the actual constructions proposed till now were not efficient. I am happy to see a paper that targets this important but often overlooked problem.
– Uses methods that are not often used in the GNN community - the paper used methods and ideas that come from the TCS community. I think it is a great addition to the toolbox of the GNN community.

Minor
– lines 160-165 could be improved. I lost the authors there.

---

> ### Author Response · Authors · 2022-08-02
> **Response to Reviewer j9Sq**
>
> Thank you very much for your valuable feedback. Below are our responses to your questions/concerns.
>
>
> **Meaning of our result to ML researchers/learnability:** Our results address the representational issues in GNNs, specifically how small GNNs can be while still being capable of distinguishing graphs according to the WL test, (which is a task that characterizes the power of GNNs, as per Xu et al; Morris et al). Although such questions do not consider learning per se, representational results have had significant impact on machine learning. (A classic example is the result due to Minsky-Papert about the limited power of single layer perceptrons, which has resulted in a decline in neural net research in the 1970s and early 1980s, until the development of deep neural networks).  The implication of our result specifically is that even very simple GNNs (such as those typically used in practice) can have the full power of the WL-test. It does not mean that such GNNs are easy to train, but that they are capable “in principle”.
>
> We also note that we are not the only ones to study this question. As mentioned in the introduction, both Xu et al and Morris et al provide intricate constructions showing that WL test can be simulated by GNNs. Our contribution is to show that this result can be achieved using much simpler GNNs than thought before.
>
>
> **The role of randomness:** The reason randomness is important is that if we fix a random seed, our algorithm will be able to simulate the WL test on most graphs, but not all graphs. However, by averaging over the random seeds, we show that for every graph, for a randomly chosen seed our algorithm works with high probability. In other words, randomness helps us ensure that our algorithm works with high probability for every graph, rather than for just the majority of graphs. This is similar to how randomness helps in a wide range of algorithms.
>
>
> **Deterministic constructions/lower bounds:** We do not know of any deterministic lower bounds beyond the ones that we prove which also hold for randomized algorithms (section 4, and appendix D in the supplementary material). Regarding deterministic upper bounds, the only ones we know of are from Xu et al. (exponential sized) and Morris et al. (polynomially sized but requires the weights of the network to depend on the graph).

---

> > ### Comment · Reviewer_j9Sq · 2022-08-07
> > **Thanks**
> >
> > Thank you for taking the time to answer my questions. I like the paper and hope it will be accepted. I agree with the authors that the expressivity results are interesting and significant for the ML community.
> > I still think the authors should connect their results to practical machine learning (perhaps in the additional page in the camera-ready version). It can be achieved by (1) formulating a model based on the theoretical findings. (2) adding a discussion and experimental results to demonstrate the validity of the theoretical observations.

---

### Meta-Review · Area_Chair_FAbQ · 2022-08-25

**Recommendation:** Accept
**Confidence:** Less certain

**Metareview:**

Even though there were some concerns regarding the limited practical consequences of these results to the ML/GNN community, the consensus is that the work makes a meaningful theoretical contribution toward the understanding of the connection between the Weisfeiler-Lehman (WL) Isomorphism Test and message-passing graph neural networks (MPNN). The work is well-written, stands out from the typical ML paper, and brings forth novel complexity bounds for the simulation of the 1WL-test with MPNN.

**Award:**

No

---

### Decision · Program_Chairs · 2022-09-14

Accept